# Anterior cingulate cortex and its input to the basolateral amygdala control innate fear response

Jinho Jhang[1], Hyoeun Lee[2], Min Soo Kang[1], Han-Sol Lee[1], Hyungju Park[2] & Jin-Hee Han[1]

Prefrontal brain areas are implicated in the control of fear behavior. However, how prefrontal circuits control fear response to innate threat is poorly understood. Here, we show that the anterior cingulate cortex (ACC) and its input to the basolateral nucleus of amygdala (BLA) contribute to innate fear response to a predator odor in mice. Optogenetic inactivation of the ACC enhances freezing response to fox urine without affecting conditioned freezing. Conversely, ACC stimulation robustly inhibits both innate and conditioned freezing. Circuit tracing and slice patch recordings demonstrate a monosynaptic glutamatergic connectivity of ACC-BLA but no or very sparse ACC input to the central amygdala. Finally, our optogenetic manipulations of the ACC-BLA projection suggest its inhibitory control of innate freezing response to predator odors. Together, our results reveal the role of the ACC and its projection to BLA in innate fear response to olfactory threat stimulus.

[1] Department of Biological Sciences, KAIST Institute for the BioCentury (KIB), Korea Advanced Institute of Science and Technology (KAIST), 291 Daehak-ro, Yuseong-gu, Daejeon 34141, Korea. [2] Department of Structure & Function of Neural Network, Korea Brain Research Institute (KBRI), 61 Cheomdan-ro, Dong-gu, Daegu 41068, Korea. Correspondence and requests for materials should be addressed to H.P. (email: phj2@kbri.re.kr) or to J.-H.H. (email: han.jinhee@kaist.ac.kr)

Fear behaviors, such as freezing in a threatening situation, are essential for survival and adaptation in humans and animals. The emotional processing system in the brain controls such fear-related behaviors in animals, and abnormalities in emotional processing contribute to fear- and anxiety-related mental disorders in humans. Multiple subdivisions in frontal lobe areas, including the medial prefrontal cortex (mPFC) and anterior cingulate cortex (ACC), play a critical role in regulating fear responses[1,2]. The prelimbic (PL) and infralimbic (IL) regions of the mPFC in rodent brains are homologous to the ventral-medial

prefrontal cortex (vmPFC) and ventral anterior cingulate cortex (vACC) in humans[1,3]. Previous studies in rodents extensively investigated the role of these two brain regions and their projections in the regulation of fear response to learned threat. The IL has been found to be a critical regulator of freezing responses after extinction[4–6]. Specifically, the activity of IL projections to the amygdala inhibits the freezing response to a conditioned threat stimulus after extinction[7]. Conversely, the PL is thought to stimulate the expression of learned fear[4]. A recent study in mice showed that inactivation of PL somata decreases the conditioned

freezing response from 80 to 40% 6 h and 7 d after conditioning[8]. Despite the accumulating evidence, however, the prefrontal circuit that controls fear response to innate threat is unknown

Research in humans and monkeys suggests that the ACC controls fear behavior in threat situations[9]. In monkeys, activation of the anterior midcingulate cortex (aMCC) is associated with changes in behavioral patterns in response to increased threat imminence[10]. Consistently, in humans, the aMCC is activated at moments of escape from virtual predators in a dynamic computer-based task[11]. Moreover, the ACC contributes to emotional regulation during fear extinction and controls emotional conflict via top-down modulation of limbic and endocrine systems[1,12,13]. In rodents, the ACC is comprised of the caudal-dorsal division of the mPFC[14–16], which is homologous to the dorsal ACC (including the aMCC) in humans[17]. Consistent with a possible role of the ACC in regulating fear behavior, anatomical analyses of rodent brains using neural circuit tracing techniques, such as injection of anterograde or retrograde tracers (e.g., fluorescent tracer dyes, horseradish peroxidase-wheat germ agglutinin (WGA), and *Phaseolus vulgaris* leucoagglutinin)[16,18–20], show that the ACC sends a strong projection to the BLA, which is involved in the regulation of fear responses to threats[21,22]. However, another line of evidence suggests an idea that the ACC is not involved in conditioned fear responses. Specifically, no significant changes in ACC activity are observed during the freezing response to a CS 1 day after contextual fear conditioning[23]. Also, conditioned freezing[23,24] is normally expressed when neuronal activity in the ACC is silenced (by lidocaine[23] and optogenetics[24]) during the retrieval of 1-day-old fear memory. Furthermore, the expression of place aversion is shown to be normal in animals with ACC lesions[25]. Despite these results suggesting that the ACC may not play a critical role in the expression of learned fear in rodents, the role of the ACC and its projection to the BLA in regulating fear responses to innate threat is unknown.

Here, we explored whether the ACC and its input to the BLA control fear responses to an innate threat in mice by employing optogenetic manipulation, bidirectional circuit tracing, and ex vivo patch-clamp recording approaches. We report that the ACC input to the BLA contributes to determine innate fear response.

## Results

**ACC photoinhibition increases an innate fear response.** As the first step for exploring a role of the caudal-dorsal area of the mPFC (covering the caudal ACC and a portion of the precentral cortex, hereafter referred to as the ACC) in regulating fear behavioral responses, we used an optogenetic silencing approach to investigate the effect of ACC inhibition on the behavioral expression of conditioned or innate fear. For ACC photoinhibition, adeno-associated viral (AAV) vectors encoding halorhodopsin-3.0 fused with enhanced yellow fluorescence protein (eNpHR3.0-eYFP; AAV-eNpHR3.0) or green fluorescence protein (eGFP; AAV-eGFP) as a control under control of the CaMKIIα promoter were injected into the bilateral ACC. A 561-nm laser light was delivered to the ACC through surgically implanted dual-fiber optic elements (Fig. 1a). Specific viral expression in the caudal ACC with no expression in the rostral ACC, PL, and IL was confirmed for each mouse after behavioral testing by microscopic observation (Fig. 1b, Supplementary Fig. 1).

Mice injected with AAV-eNpHR3.0 or AAV-eGFP were trained for auditory fear conditioning in which an auditory tone as a CS was paired with a foot shock as an unconditioned stimulus (US), and 24 h later mice were re-exposed to the tone CS in a different context (Fig. 1c). During testing, mice received three light off/on cycles in the presence of the tone CS after a pre-CS period. Mice in both groups exhibited freezing responses to the tone CS that were not significantly changed by light stimulation (Fig. 1d–f), suggesting that inhibition of ACC activity does not affect the expression of conditioned fear behavior to an auditory stimulus. This result is consistent with previous reports showing that inhibition of ACC activity does not affect the freezing response to the CS after fear conditioning[23,24].

Fear behavior is provoked in response to not only learned threats but also innate threats such as a predator-related sensory stimulus or a live predator in prey animals[26]. Thus, we examined whether ACC inhibition affects the freezing response to an innate threat. Because exposure to natural fox urine or its ingredients elicits fear behavior and a physiological stress response in rodents[27–29], we used fox urine as an innate threat stimulus. We first confirmed that mice exhibited freezing in the presence of fox urine (Fig. 1g, h). Consistent with fear response to the fox urine, we also found that the fox urine exposure induced the activation of the dorsal periaqueductal gray, a brain structure critical for innate fear responses[21,28,30–32] (Supplementary Fig. 2). In optogenetic behavior experiment, mice were placed in a cage with fox urine and received three light off/on cycles (Fig. 1i). Unlike auditory fear conditioning, we here observed significantly increased freezing in eNpHR3.0-expressing mice during light-on periods compared with light-off periods in a reversible manner (Fig. 1j), whereas eGFP-expressing mice exhibited no significant

**Fig. 1** Photoinhibition of CaMKIIα-positive neurons in the ACC increases the freezing response to an innate threat but not to a learned threat. **a** Illustration of eNpHR3.0-based silencing of CaMKIIα-positive neurons in the ACC. **b** Left, representative confocal image of eNpHR3.0-eYFP expression in the ACC. Scale bar, 500 μm. Right, illustration of viral injection into the ACC. **c** Experimental procedure for auditory fear conditioning test with photoinhibition of ACC neurons. **d, e** Time spent freezing during the retention test was not affected by photoinhibition in the eNpHR3.0 ($n = 7$) (**d**) or eGFP ($n = 7$) (**e**) groups. Paired *t*-tests for eNpHR3.0 ($P = 0.1699, = 0.9338, = 0.0991$) and eGFP ($P = 0.4979, = 0.3872, = 0.4103$). **f** Light-induced change in percentage of freezing behavior for **d** and **e**, calculated by subtracting freezing values within the first OFF–ON cycle. Unpaired *t*-test ($P = 0.4962, t = 0.7018,$ df = 12). **g** Illustration of fox urine exposure test (empty cage or water for control groups). **h** Left, time spent freezing during the exposure test in empty cage ($n = 4$), water ($n = 4$), and fox urine ($n = 4$) groups. Repeated measures two-way ANOVA ($F_{(6, 27)} = 1.663, P = 0.1687$) and Sidak's post hoc test (empty cage vs. fox urine, $P = 0.3883, = 0.0144, = 0.0028, = 0.0002$; water vs. fox urine, $P = 0.5258, = 0.0653, = 0.2224, = 0.0002$). Right, freezing behavior during the 3–4 min period. One-way ANOVA ($F_{(2, 9)} = 2.119, P = 0.0062$) and Tukey's post hoc test (empty cage vs. fox urine, $P = 0.0108$; water vs. fox urine, $P = 0.0118$). **i** Experimental procedure for fox urine test with photoinhibition of ACC neurons. **j, k** Time spent freezing during the fox urine test was selectively increased by photoinhibition in the eNpHR3.0 group ($n = 7$) (**j**) but not in the eGFP group ($n = 8$) (**k**). Paired *t*-tests for eNpHR3.0 ($P = 0.0152, = 0.0001, = 0.1520$) and eGFP ($P = 0.4352, = 0.7963, = 0.9610$). **l** Light-induced change in percentage of freezing behavior for **j** and **k**, calculated by subtracting freezing values within the first OFF–ON cycle. Unpaired *t*-test ($P = 0.0049, t = 3.384,$ df = 13). **m** Experimental procedure for water control testing with photoinhibition of ACC neurons. **n** Time spent freezing during the water control test was not affected by light stimulation ($n = 5$). Paired *t*-tests ($P = 0.7610, = 0.2585, = 0.2609$). Line and bar graphs, mean ± standard error of the mean (s.e.m.). Box-whisker plots, median and interquartile range with 5-95 percentile distribution. Independent groups of mice were used for **d–f** and **j–l**. *$P < 0.05$, **$P < 0.01$, ***$P < 0.001$

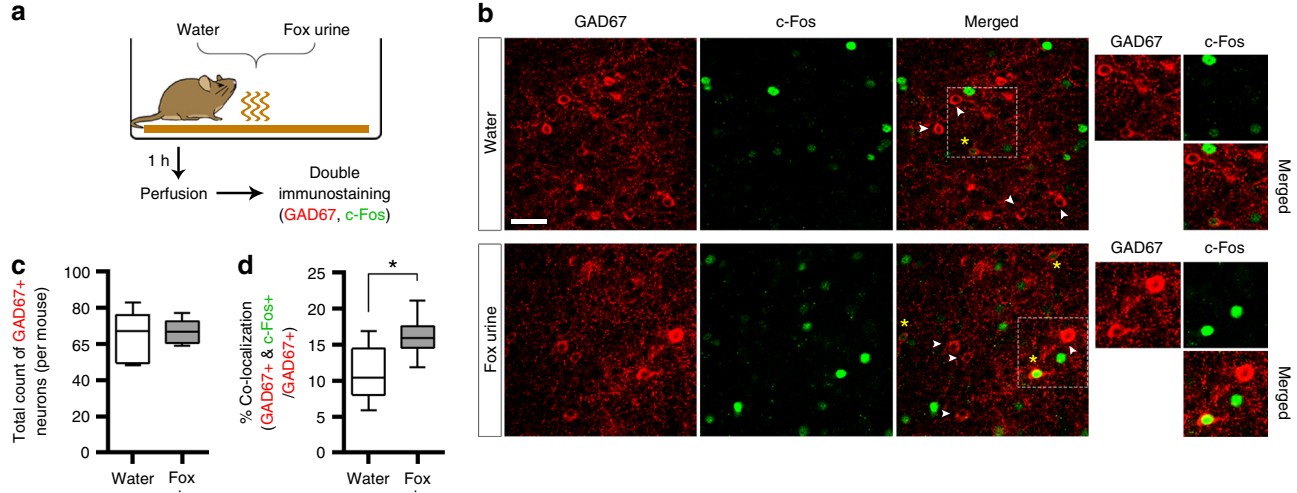

**Fig. 2** GABAergic interneurons in the ACC are activated by exposure to fox urine. **a** Experimental procedure for double immunostaining experiment (GAD67 and c-Fos). Mice were perfused after 1-h exposure to water ($n = 5$; control group) or fox urine ($n = 6$). **b** Representative confocal images showing GAD67+(red) and c-Fos+(green) signals observed in the ACC field. Wedge, GAD67+cell but no c-Fos signal. Yellow asterisk, GAD67 & c-Fos double+cell. Right images are magnification of dashed square area. Scale bar, 40 μm. **c** Total count of GAD67+cells included for the quantitative analysis. Unpaired t-test ($P = 0.6313$, $t = 0.4967$, df = 9). **d** The percentage of GAD67+cells that are co-localized with c-Fos+signals. Unpaired t-test ($P = 0.0388$, $t = 2.417$, df = 9). Box-whisker plots are expressed as median, interquartile range with 5-95 percentile distribution. *$P < 0.05$

changes in freezing during the test (Fig. 1k). A pairwise comparison confirmed a significant group difference in light-induced change in freezing (Fig. 1l). These results can be interpreted that silencing ACC activity itself may non-specifically induce freezing. To test this possibility, we performed the same behavioral test in the presence of water instead of fox urine (Fig. 1m). In this condition, light stimulation did not induce freezing (Fig. 1n), thus excluding the above possibility. Moreover, we found no changes in the locomotor and anxiety behaviors by light in open field test (Supplementary Fig. 3a–c). Taken together, these results suggest that ACC inhibits freezing response to an innate threat.

Given this finding, it is possible that ACC activity may be reduced during innate fear expression to promote freezing response to the fox urine perhaps by the activation of local inhibitory interneurons. We tested this idea by performing a double immunostaining analysis for c-fos and glutamic acid decarboxylase (GAD67) in the water and fox urine condition (Fig. 2a). The number of GAD67+ and double+cells (Fig. 2b) was quantified in a double-blinded manner (see methods). We found a significant increase of the percentage number of c-Fos +cells in GAD67+cell population (% co-localization) in mice exposed to the fox urine compared to the control mice exposed to water (Fig. 2c, d). These results are thus consistent with the above idea, and support our conclusion that ACC inhibits freezing response to an innate threat.

**ACC photoactivation reduces the innate fear response**. Considering that silencing ACC activity increases innate freezing, we predicted that artificial activation of the ACC may reduce the freezing response to fox urine. To test this idea, we optogenetically activated CaMKIIα-positive neurons in the ACC using Channelrhodopsin-2 (ChR2). For ACC photoactivation, AAV vectors encoding ChR2 fused with Venus (AAV-ChR2) or eGFP (AAV-eGFP), both of which expression were driven by the CaMKIIα promoter, were injected into the bilateral ACC, and 473-nm laser light was delivered to the ACC through implanted dual-fiber elements (Fig. 3a and Supplementary Fig. 4). During testing, mice were placed in a cage containing fox urine, and their

freezing behavior was monitored using the same optogenetic manipulation procedures as Fig. 1 (Fig. 3b). When the light was turned off, mice exhibited modest levels of freezing, but ChR2-expressing mice exhibited significantly reduced freezing in a reversible manner over repeated cycles during the light-on periods (Fig. 3c). By contrast, eGFP-expressing mice showed no significant changes in freezing during the test (Fig. 3d). A pairwise comparison confirmed a significant group difference in light-induced change in freezing (Fig. 3e). We found no changes in the locomotor and anxiety behaviors by the photoactivation in open field test (Supplementary Fig. 3d and 3e). These results indicate that increased activity of CaMKIIα-positive neurons in the ACC inhibits fear response to predator odor, further supporting that ACC inhibits innate freezing response.

**ACC photoactivation reduces a conditioned fear response**. Consistent with previous studies[23,24], our data showed that conditioned freezing is not affected by ACC inactivation (Fig. 1d, f). However, it is possible that artificial activation of the ACC affects the expression of conditioned freezing. To test this possibility, mice were injected with AAV-ChR2 or AAV-eGFP and trained for contextual fear conditioning followed by fear memory test 24 h later. During testing, mice were returned to the conditioned context and received three light off/on cycles to activate the ACC (Fig. 4a, b). Before light stimulation, mice in both groups displayed strong freezing in the conditioned context. However, ChR2-expressing mice displayed a robust reduction in freezing when the light was turned on in a reversible manner over repeated cycles (Fig. 4c). By contrast, eGFP-expressing mice did not show a light-induced alteration of freezing response during the test (Fig. 4d). A pairwise comparison confirmed a significant group difference in light-induced change in freezing (Fig. 4e).

Next, groups of mice that received AAV-ChR2 or AAV-eGFP injection were trained for auditory fear conditioning and tested 24 h later. During testing, mice received three light off/on cycles in the presence of the tone CS (Fig. 4f). As expected, mice exhibited robust freezing after CS onset compared with pre-CS baseline. Consistent with our contextual fear conditioning results, ChR2-expressing mice showed a significant reduction in

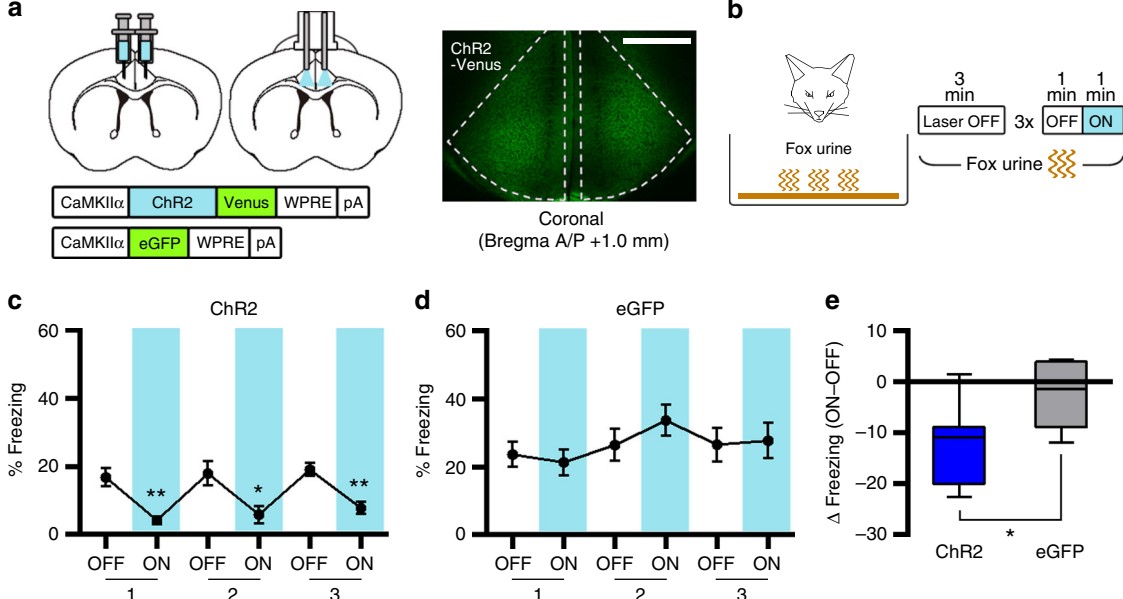

**Fig. 3** Photoactivation of CaMKIIα-positive neurons in the ACC inhibits the freezing response to an innate threat. **a** Left, illustration of ChR2-based activation of CaMKIIα-positive neurons in the ACC. Right, representative confocal image (coronal slice; A/P+1.0 mm) of ChR2-Venus expression in the ACC. Scale bar, 500 μm. **b** Experimental procedure for fox urine test with photoactivation of ACC neurons. **c**, **d** Percentage of time spent freezing during the fox urine test was selectively reduced by photoactivation in the ChR2 group ($n = 7$) (**c**) but not in the eGFP group ($n = 6$) (**d**). Paired $t$-tests for ChR2 ($P = 0.0064, = 0.0280, = 0.0058$) and eGFP ($P = 0.4538, = 0.3552, = 0.8933$). **e** Light-induced change in percentage of freezing behavior, calculated by subtracting freezing values within the first OFF–ON cycle, for ChR2 (**c**) and eGFP (**d**) groups. Unpaired $t$-test ($P = 0.0360, t = 2.387$, df = 11). Line graphs are expressed as mean ± s.e.m. Box-whisker plot is expressed as median, interquartile range with 5-95 percentile distribution. *$P < 0.05$, **$P < 0.01$

conditioned freezing to the tone CS in a reversible manner over repeated cycles (Fig. 4g). The eGFP-expressing control mice displayed no significant changes in freezing during the test (Fig. 4h). A pairwise comparison confirmed a significant group difference in light-induced change in freezing (Fig. 4i). Taken together, our results indicate that ACC activation is sufficient to inhibit conditioned fear responses.

**ACC projection to the BLA is monosynaptic and glutamatergic.** As the amygdala is a key brain region controlling fear responses to both innate and learned threats[28], and previous studies show a strong ACC projection to the amygdala in rodents[16,20], we reasoned that an ACC projection to the amygdala may mediate the inhibition of innate fear behavior. To test this idea, we first determined the sub-region of the amygdala to which the ACC sends a major output projection. The ACC projection terminals in the amygdala were visualized by the expression of synaptobrevin-2 (Syb2, a presynaptic marker[33]) in the ACC neurons. AAV vector (AAV-mCh-IRES-eGFP-Syb2) carrying the gene of eGFP fused-Syb2 with mCherry-IRES[34] was injected into the bilateral ACC (Fig. 5a, 3 mice), and 6 weeks later we visualized eGFP signals in the amygdala with a confocal microscope. Correct placement of the virus injection sites in the ACC was confirmed by mCherry expression (Fig. 5a). We observed strong eGFP signals in the BLA consistently across three different mice (Fig. 5b, c), indicating that the BLA is the major ACC projection target sub-nucleus in the amygdala. Previous tracing studies also report similar ACC projection pattern with our data, strong projection to the BLA but no or very few in the lateral nucleus (LA)[16,20]. Next, to identify the ACC neurons that project to the amygdala, we performed a retrograde tracing experiment using canine adenovirus-2 (CAV2), which is retrogradely transferred to cell somata via axonal retrograde transport[35]. CAV2 encoding Cre-recombinase (CAV2-Cre) was injected into the BLA in the right hemisphere, and the AAV encoding a Cre-dependent eYFP

expression cassette (AAV-EF1α-DIO-eYFP) was injected into the bilateral ACC (5 mice; Fig. 5d). Confocal imaging showed that neurons with strong eYFP signals were mostly found in layer V of the ipsilateral ACC, whereas fewer eYFP signals were found in the contralateral ACC (Fig. 5e). Quantification analysis confirmed such ipsilateral connectivity pattern in all five mice used for the tracing experiment (Fig. 5f), suggesting that BLA-projecting ACC neurons are mainly located in layer V of the ipsilateral ACC.

We next investigated whether the ACC-BLA synapses are excitatory or inhibitory by performing immunohistochemical profiling. Sections of the amygdala expressing eGFP-Syb2 were double-immunostained with antibodies against VGLUT1 and VGAT, which are enriched at glutamate- and GABA-containing synaptic vesicles, respectively[36,37]. Confocal imaging showed that the majority (89.2%) of immunostaining signals co-localized with eGFP-Syb2 puncta were VGLUT1-positive (Fig. 5g, h), whereas the minority (8.2%) were VGAT-positive. Our data thus suggest that ACC-BLA synapses are glutamatergic.

To further investigate the nature of the ACC-BLA synaptic transmission, we next recorded ACC-BLA synaptic activity by performing ex vivo whole-cell patch-clamp recording (Fig. 5i). We adopted a Cre-dependent anterograde transneuronal labeling strategy using serotype 1-packaged AAV (AAV1)[38]. Two different AAV constructs (AAV1-hSyn-Cre and AAV2-DIO-ChR2-eYFP) were co-injected into the bilateral ACC of a transgenic mouse line (Ai9) carrying floxed-stop-tdTomato. In this experimental design, ChR2 is supposed to be expressed only in the neurons co-expressing Cre in the ACC, and the postsynaptic target neurons of the ACC, if any, would be labelled with tdTomato by the anterograde transfer of AAV1-hSyn-Cre followed by Cre-induced excision of the STOP sequence from floxed-STOP-tdTomato. Four weeks after the injection, we indeed observed tdTomato-positive (tdT+) cells in the BLA (Fig. 5j). We then made a whole-cell patch clamp onto the tdT+BLA neurons and recorded the PSP responses induced by light (470 nm)

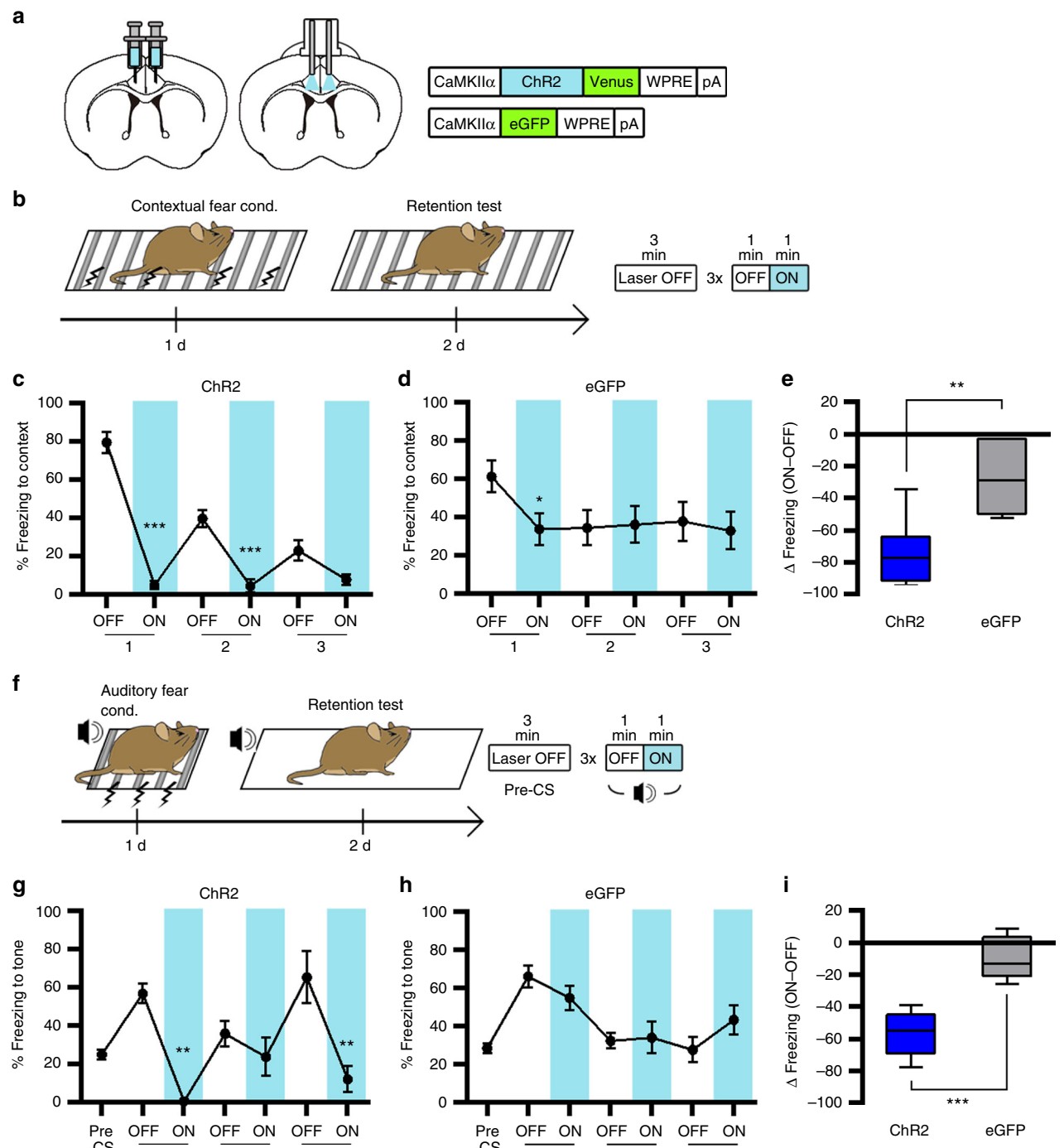

**Fig. 4** Photoactivation of CaMKIIα-positive neurons in the ACC inhibits freezing responses to conditioned stimulus. **a** Illustration of ChR2-based activation of CaMKIIα-positive neurons in the ACC. **b** Experimental procedure for contextual fear conditioning test with photoactivation of ACC neurons. **c**, **d** Percentage of time spent freezing during the contextual fear memory test was reduced by photoactivation in the ChR2 group (*n* = 8) (**c**). In the eGFP group (*n* = 6) (**d**), percentage of time spent freezing was reduced during the first OFF–ON cycle but not the second and third cycles. Paired *t*-tests for ChR2 (*P* < 0.0001, < 0.0001, = 0.0634) and eGFP (*P* = 0.0261, = 0.7848, = 0.3823). **e** Light-induced change in percentage of freezing behavior, calculated by subtracting freezing values within the first OFF–ON cycle, for ChR2 (**c**) and eGFP (**d**) groups. Unpaired *t*-test (*P* = 0.0011, *t* = 4.276, df = 12).
**f** Experimental procedure for auditory fear conditioning test with photoactivation of ACC neurons. **g**, **h** Percentage of time spent freezing during the auditory fear memory test was selectively reduced by photoactivation in the ChR2 group (*n* = 9) (**g**) but not in the eGFP group (*n* = 7) (**h**). Wilcoxon matched-pairs rank tests for ChR2 (*P* = 0.0039, = 0.3594, = 0.0039) and paired *t*-tests for eGFP (*P* = 0.0617, = 0.7975, = 0.1031). **i** Light-induced change in percentage of freezing behavior, calculated by subtracting freezing values within the first OFF–ON cycle, for ChR2 (**g**) and eGFP (**h**) groups. Unpaired *t*-test (*P* < 0.0001, *t* = 6.616, df = 14). Line graphs are expressed as mean ± s.e.m. Box-whisker plots are expressed as median, interquartile range with 5-95 percentile distribution. *\*P* < 0.05, **\*\*P* < 0.01, ***\*\*\*P* < 0.001

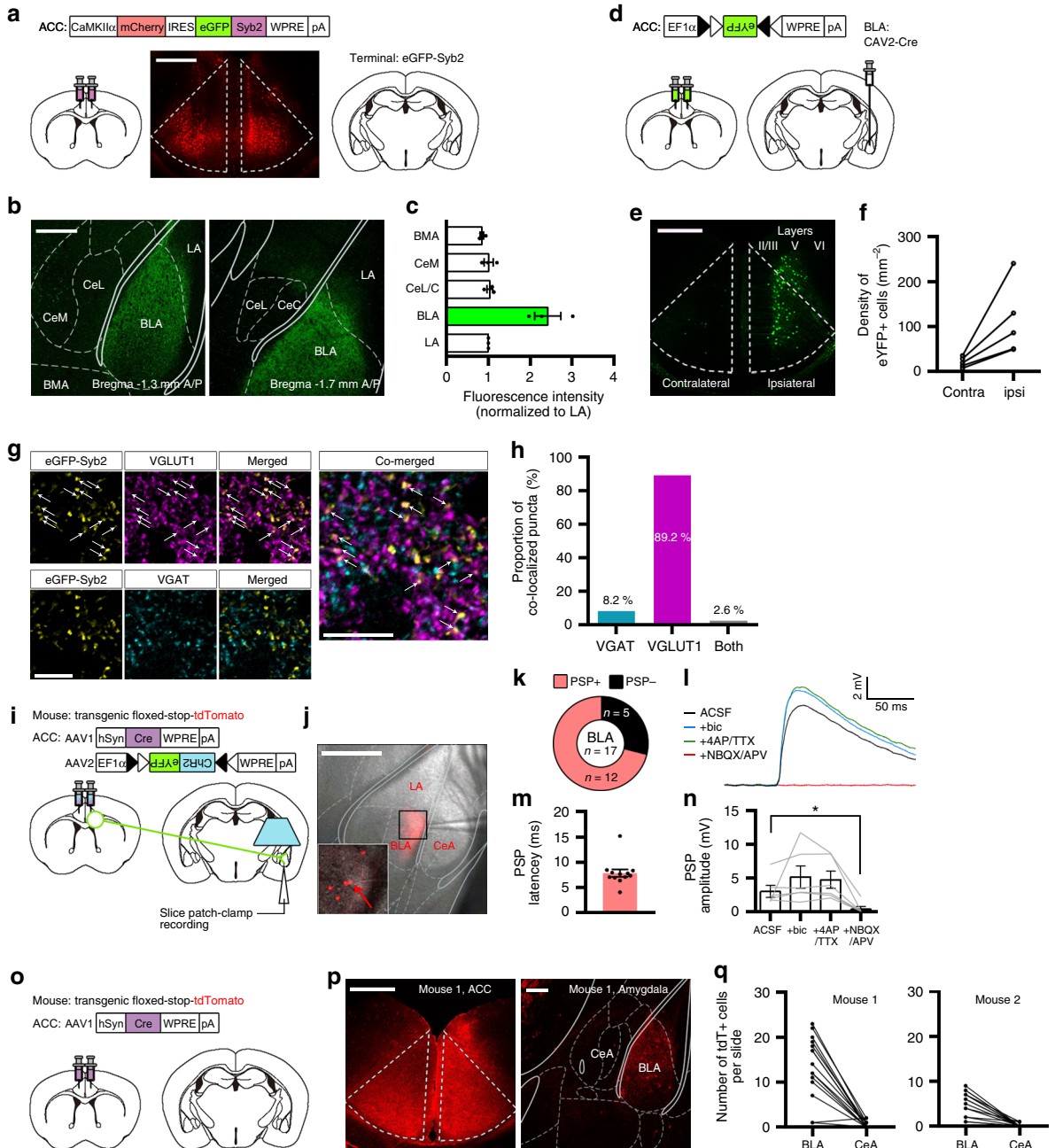

**Fig. 5** ACC projection to the BLA is monosynaptic and glutamatergic. **a** Anterograde terminal tracing strategy based on expression of eGFP-Syb2. Confocal image shows mCherry-expressing cells selectively observed in the ACC. Scale bar, 500 μm. **b** Representative confocal images showing eGFP-Syb2 expression pattern in the amygdala sub-nuclei. Strong eGFP-Syb2 expression was observed in the BLA. Scale bar, 200 μm. **c** Quantitative analysis of the eGFP-Syb2 signals in the amygdala sub-nuclei (across three different mice). Fluorescence intensity was normalized to mean intensity value (0–255 scale) measured in the LA area. **d** Retrograde tracing strategy using CAV2-Cre-dependent expression of DIO-eYFP. **e** Representative confocal image showing eYFP-expressing cells in the ACC. Scale bar, 500 μm. **f** Quantitative analysis of the retrogradely labeled neurons (eYFP+) in the ipsilateral and contralateral ACC. **g** Representative confocal images showing VGLUT1 or VGAT immunoreactive signals with eGFP-Syb2 signals. Scale bar, 10 μm. **h** Bar graph showing co-localized puncta in the BLA. **i** Illustration of ex vivo patch-clamp recording experiments for monitoring light-evoked PSPs in ACC input-identified BLA neurons. **j** Representative epi-fluorescence image indicating the patch-clamp recording site. Bottom left image, magnification of the black square area. tdTomato-positive (tdT+) cells were selectively targeted, as indicated by red arrow. Scale bar, 500 μm. **k** Proportion of neurons displaying PSP responses to light stimulation. Total 17 tdT+BLA neurons were successful for whole-cell patch-clamp recording from 10 slices of 5 mice. 12 of 17 cells were PSP-positive (PSP+). **l** Representative recording traces indicating light-evoked PSPs in the ACSF (black), and in the presence of bicuculline (+bic; blue), bicuculline+4-AP+TTX (+4AP/TTX; green), or bicuculline+4-AP+TTX+NBQX+APV (+NBQX/APV; red). **m** PSP latency of the PSP+/tdT+BLA neurons recorded ($n = 12$ cells). **n** Summary of amplitudes of light-evoked PSPs measured from each drug treatment condition ($n = 6$ cells from 6 slices of 3 mice). Repeated measures one-way ANOVA ($F_{(1.167, 5.837)} = 6.900$, $P = 0.0376$) with Dunnett's post hoc test (ACSF vs.+NBQX/APV; $P = 0.0106$). Data are expressed as mean ± s.e.m. *$P < 0.05$. **o** Anterograde tracing strategy based on transneuronal delivery of AAV1-hSyn-Cre. **p** Representative confocal images showing the pattern of tdT+expression in the ACC (left, scale bar: 500 μm) and the amygdala (right, scale bar: 200 μm). **q** Quantitative analysis of the tdT+neurons in the BLA and CeA. BMA, basomedial nucleus; LA, lateral nucleus; CeA (CeM, CeL, and CeC), central nucleus (medial, lateral and capsular subdivisions) of the amygdala[15]

illumination to stimulate ChR2-expressing ACC axon terminals. Our results showed that 12 of 17 (71%) tdT+BLA cells (from 10 slices of 5 mice) displayed successful PSP responses to the light stimulation (Fig. 5k–m). The latency of light-evoked PSPs was $6.4 \pm 0.7$ ms ($n = 12$), which is in the range of monosynaptic events (Fig. 5m). To confirm whether light-evoked PSPs recorded from BLA cells are either monosynaptic or glutamtergic, we sequentially treated the slices with inhibitors such as bicuculline, tetrodotoxin (TTX)+4-aminopyridine (4-AP), and 2,3-dihydroxy-6-nitro-7-sulfamoyl-benzo[f]quinoxaline-2,3-dione (NBQX)+D-2-amino-5-phosphonopentanoate (D-APV), and measured the effect of such drugs on light-evoked PSPs. We were able to maintain stable light-evoked PSP recordings from 6 of 12 cells during sequential drug treatments. Although there was no statistical significance, we observed a tendency that bicuculline application slightly enhanced PSP amplitudes, which were not affected by the additional treatment of 4-AP/TTX (PSP amplitude in mV: ACSF = $3.0 \pm 0.9$, bicuculline = $5.1 \pm 1.7$, 4-AP/TTX = $4.7 \pm 1.3$; Fig. 5l, n), indicating that light-evoked PSPs are slightly regulated by unknown GABAergic inputs and mediated by monosynaptic events. These PSPs were completely abolished when both NBQX and APV were included in the solution (PSP amplitude in mV: NBQX+APV = $0.4 \pm 0.4$; Fig. 5l, n). We replicated these results by using age-matched 129 × C57Bl/6 wildtype mice only injected with AAV-CaMKIIα-ChR2-Venus in the ACC (Supplementary Fig. 5a–e): our data showed that randomly selected BLA cells (11 of 14 cells; 79%) showed light-evoked PSPs, which was not detected from all selected CeA cells (11 cells). Light-evoked PSPs from BLA cells were increased by a treatment with bicuculline but completely diminished by co-application of NBQX and APV (Supplementary Fig. 5c and 5e). These data indicate that the ACC-BLA synapses are monosynaptic and glutamatergic.

Using the same approach, we performed an anterograde tracing experiment to further examine the ACC projection pattern in the amygdala (Fig. 5o). Two Ai9 mice were injected with the AAV1 anterograde tracer (high titer AAV1-hSyn-Cre), and imaging was performed 8 weeks later to ensure the sufficient expression of tdTomato. We collected consecutive brain sections covering the entire BLA and CeA area (A/P coordinate −1.0 to −1.9 mm), and compared the number of tdT+cells in the BLA and CeA at each slide (Fig. 5p). We observed a predominant expression of tdT+cells in the BLA (total 204 in mouse 1 and 77 cells in mouse 2, respectively; Fig. 5q). In the CeA, very few tdT+cells were detected (total 8 in mouse 1 and 3 cells in mouse 2, respectively). These results are consistent with our eGFP-Syb2-tracing data above (Fig. 5a–c) and ex vivo patch-clamp recording results (Fig. 5i–n), indicating that there is a major ACC input to the BLA and very sparse input to the CeA.

**Selective photoactivation of ACC neurons projecting to BLA.** If the ACC-BLA projection is critical for the control of innate fear behavior, the selective activation of ACC neurons projecting to the BLA should reduce the freezing response to fox urine. To test this possibility, retrograde tracing viral vector, CAV2-Cre, was injected into the bilateral BLA, and Cre-dependent AAV constructs (DIO-ChR2-eYFP or DIO-eYFP) were injected into the bilateral ACC (Fig. 6a). Consistent with our tracing data (Fig. 5), confocal imaging results showed major expression of ChR2-eYFP or eYFP in neurons in ACC layer V (Fig. 6b). For light delivery, dual-fiber optic elements were implanted into the bilateral ACC. During behavioral testing, mice were placed in a cage containing fox urine, and freezing behavior was monitored over three light off/on cycles (Fig. 6c). ChR2-expressing mice showed a light-induced reduction in freezing during the first and second light-on

periods (Fig. 6d), similar to that observed after bulk ACC activation, whereas eYFP-expressing mice showed no significant changes in freezing (Fig. 6e). A pairwise comparison confirmed a significant group difference in light-induced change in freezing (Fig. 6f). These results suggest that ACC projection to the BLA is a critical circuit pathway for regulating innate fear response to fox urine.

**ACC-BLA projection controls the innate fear response.** To determine whether the ACC-BLA projection controls innate fear response to fox urine, we next employed projection-based optogenetic approaches. By using AAV vectors, we expressed eNpHR3.0-eYFP or eGFP in CaMKIIα-positive neurons in the bilateral ACC and delivered the light through fiber-optic elements implanted in the bilateral amygdala by using the coordinate of ML ±3.2 mm (Supplementary Fig. 6a). During behavioral test, mice were placed in a cage containing fox urine, and freezing behavior was monitored over three light off/on cycles. We found that eNpHR3.0-expressing mice exhibited a significantly higher level of freezing during the light-on periods than that during the light-off periods in a reversible manner over repeated cycles (Supplementary Fig. 6b). By contrast, eGFP-expressing mice showed no significant changes in freezing during the test (Supplementary Fig. 6c). A pairwise comparison confirmed a significant group difference in light-induced change in freezing (Supplementary Fig. 6d). Because the placement of optic fiber was biased to the CeA in this condition (Supplementary Fig. 7a), we repeated the same experiments but with altered fiber-implantation coordinate covering both BLA and CeA areas (M/L ±3.35 mm, Fig. 7a–c and Supplementary Fig. 7b). Same light-induced behavioral alterations (as in Supplementary Fig. 6) were observed in this condition (Fig. 7d, e). These results thus support that ACC-BLA projection controls innate freezing response to fox urine.

Because the PL cortex is located close to the ACC and also has a projection to the BLA[18,20], it is possible that freezing behavior change by our optogenetic manipulation was caused by unintended manipulation of PL-BLA projection. To address this issue, we also examined whether the PL-BLA photoinhibtion induces the increase of freezing behavior to fox urine (Fig. 7f). A different coordinate corresponding to the PL was used for the injection of AAV-eNpHR3.0, while the coordinate for fiber-optic implant was identical to the ACC-BLA experiments. Unlike the ACC-projection eNpHR3.0 group (Fig. 7d), photoinhibition of PL projection did not induce increased freezing response rather there was a tendency of decrease in freezing (Fig. 7f, g), thus confirming the specificity of our manipulation of ACC projection.

Next, to determine whether activation of ACC-BLA projection is sufficient to inhibit the innate fear response, we directly stimulated the ACC terminals in the BLA by injecting AAV vectors expressing CaMKIIα promoter-driven ChR2-Venus (or eGFP control) in the bilateral ACC (Fig. 7h) and delivering light pulses to the BLA area through implanted fiber elements (Supplementary Fig. 7c). Our results showed that ChR2-expressing mice displayed reduced freezing behavior during the light-on periods of fox urine test (Fig. 7i), whereas no significant changes were observed in the eGFP control mice (Fig. 7j, k). We found no changes in the locomotor and anxiety behaviors by the photoactivation of ACC-BLA projection (Supplementary Fig. 8), indicating a specific regulation of innate fear behavior by activation of ACC-BLA projection. The efficient optogenetic manipulation of ACC-BLA projection in our condition was confirmed by ex vivo patch-clamp recordings. We recorded light-evoked PSPs from BLA neurons of slices with expression of CaMKIIα-driven ChR2-Venus in the ACC, and the light was

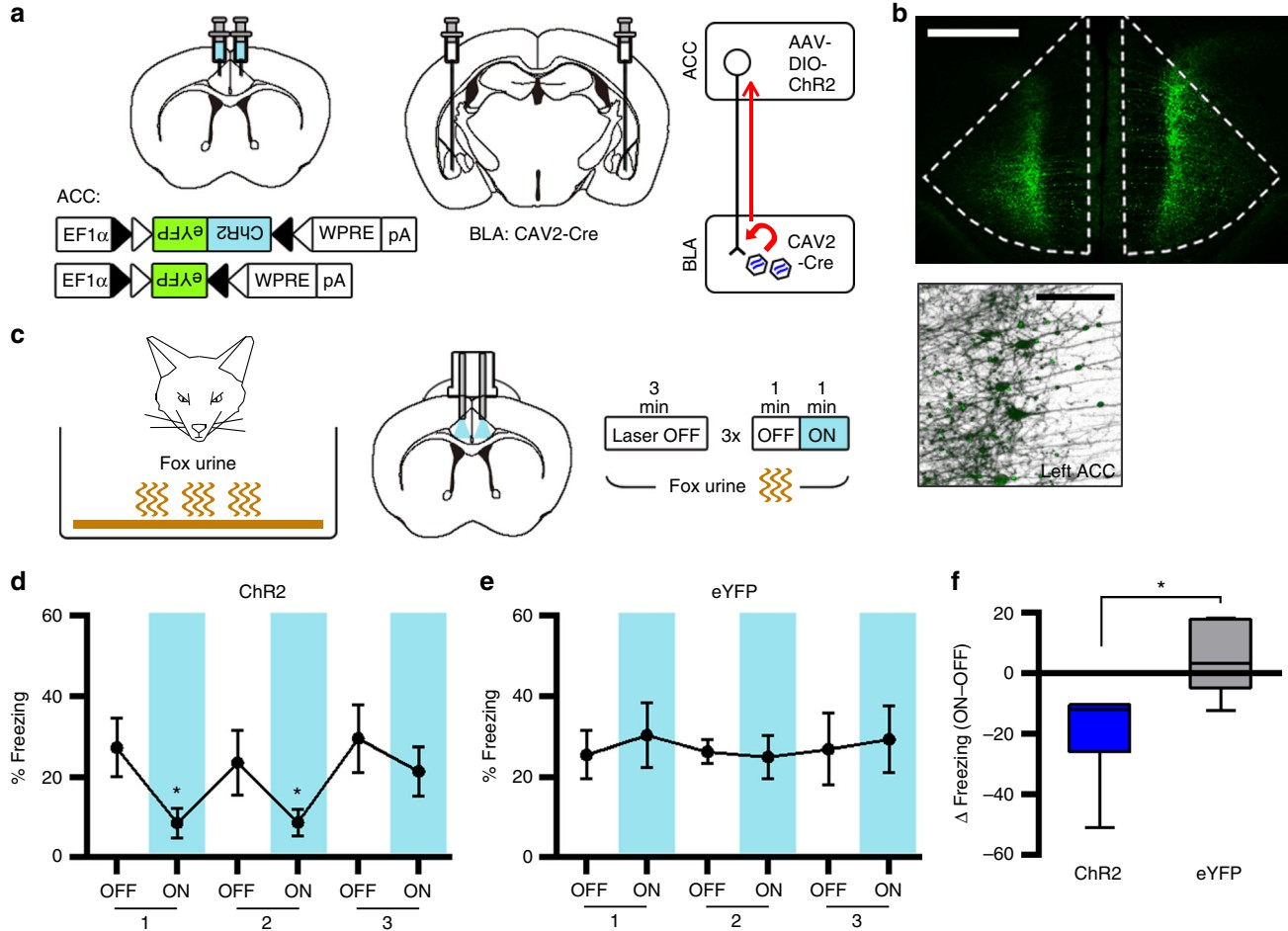

**Fig. 6** Photoactivation of ACC neurons projecting to the BLA inhibits the freezing response to an innate threat. **a** Illustration of retrograde labeling strategy of ACC neurons projecting to the BLA with CAV2-Cre-dependent expression of AAV-DIO-ChR2-eYFP. **b** Top, representative confocal microscopic image showing the expression of DIO-ChR2 in ACC layer V neurons. Bottom, image of 3-D reconstructed neurons (by serial Z-stack confocal imaging) observed in the left ACC area. Scale bar, 500 μm (top) and 100 μm (bottom). **c** Experimental procedure for fox urine test with photoactivation of ACC neurons projecting to the BLA. **d**, **e** Percentage of time spent freezing during the fox urine test was selectively reduced by photoactivation in the ChR2 group ($n = 6$) (**d**) but not in the eYFP group ($n = 6$) (**e**). Wilcoxon matched-pairs rank test for ChR2 ($P = 0.0313, = 0.0313, = 0.2188$) and eYFP ($P = 0.3125, = 0.5625, > 0.9999$). **f** Light-induced change in percentage of freezing behavior, calculated by subtracting freezing values within the first OFF–ON cycle, for ChR2 (**d**) and eGFP (**e**) groups. Mann–Whitney U test ($P = 0.0152$). Line graphs are expressed as mean ± s.e.m. Box-whisker plot is expressed as median, interquartile range with 5-95 percentile distribution. *$P < 0.05$

delivered to the ACC terminal with similar parameters in animal behavior tests (Supplementary Fig. 5f–i). The BLA cells showed stable PSP responses to 10 Hz light stimulation with a relatively high PSP-reliability (4 of 5 cells with 100% PSP-reliability; Supplementary Fig. 5g and 5h). None of CeA cells displayed detectable PSP responses in the same slice (Supplementary Fig. 5g and 5i). Taken together, these results support that the ACC-BLA projection inhibits innate freezing response.

**Specificity of the control function of ACC-BLA projection**. To investigate the specificity of the fear response control by ACC-BLA circuit pathway, we first determined whether photoinhibition of ACC-BLA projection affects avoidance response to the fox urine (Supplementary Fig. 9). For this experiment, we used a method described in previous study in rats[29,39] (see methods). In brief, we placed a petri dish containing either fox urine or water as a control at one corner (sample quadrant) and measured the time duration mice spent at that sample quadrant (Supplementary Fig. 9a). Consistent with the previous report in rats, we also observed a similar avoidance behavior to the fox urine in wild-type mice compared to water control, validating our behavioral

test set-up (Supplementary Fig. 9b). Next we performed the same behavioral test with mice expressing eNpHR3.0 (or eGFP control) in ACC-BLA projection to examine the effect of inhibiting ACC-BLA projection on the avoidance response to fox urine (Supplementary Fig. 9c). In contrast to alteration of the innate freezing behavior, we found no significant difference in the avoidance behavior between the eNpHR3.0 and eGFP (control) groups in both light off and on session (Supplementary Fig. 9e). These results suggest that the regulatory role of ACC-BLA projection is specific to the freezing behavior. Nevertheless, our results do not completely rule out the possibility that ACC-BLA projection may be involved in regulating other defensive behaviors such as risk assessment[2,26].

Next, we examined whether ACC-BLA projection controls fear response to other predator odors. For this purpose, we tested the effect of optogenetic inactivation of the ACC-BLA projection on freezing response to other predator urines. We first tested the freezing response to cat urine (Fig. 8a). The eNpHR3.0 was expressed in the bilateral ACC, and ACC-BLA projection was inhibited by the 560-nm light illumination in the BLA as before. In the presence of cat urine (10 ml of 100% wild cat urine

(PeteRikards)), mice exhibited no significant change in freezing response with the photoinhibition (Fig. 8b–d). However, when coyote urine (10 ml of 100% coyote urine (Leg Up Enterprise)) was adopted as another predator odor to cause innate fear response (Fig. 8e), mice displayed a significant increase in freezing response with photoinhibition of ACC-BLA projection compared to eGFP control group (Fig. 8f–h). According to

previous reports, cat urine may induce no or very weak fear response including freezing in rodents[29,40]. Thus, it is likely that our optogenetic manipulation affected freezing elicited by coyote but not cat urine because cat urine does not provoke fear strongly. Nevertheless, because both cat and coyote urine provoked similar level of freezing at around 15–20%, our results do not exclude the possibility that ACC-BLA projection does not control innate

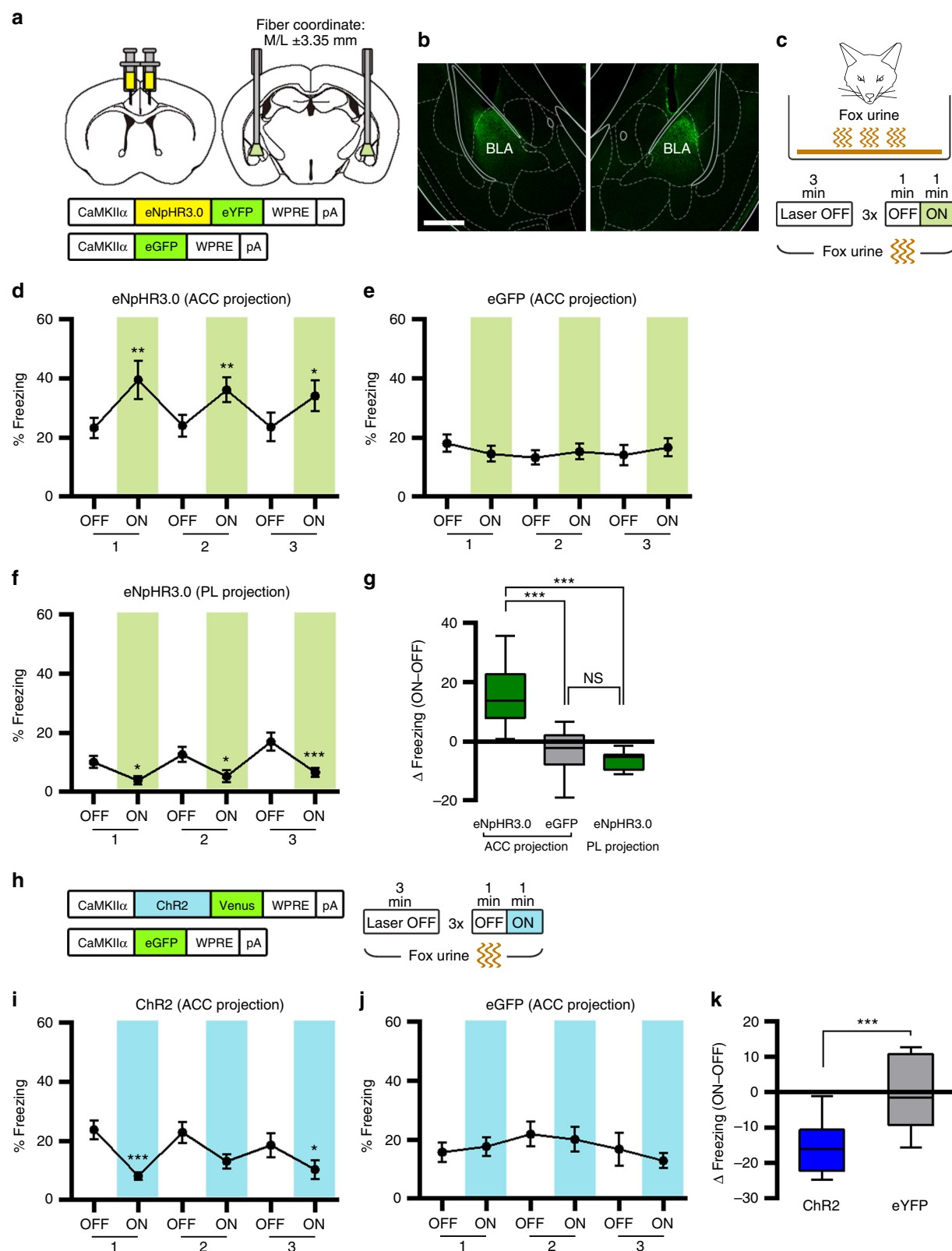

freezing response to cat urine. Together, these results support that ACC-BLA projection has a general control function for the freezing response to the predator odor, but not specific to fox urine.

Finally, we examined whether ACC-BLA projection controls innate freezing response to a live predator such as a rat. When we exposed mice to the rat (Fig. 8i), mice exhibited a moderate level of freezing (18.7 ± 3.1%) but showed enhanced freezing responses by optogenetic inhibition of ACC-BLA projection (Fig. 8j–l). Taken together, these results thus suggest that the ACC-BLA projection is involved in regulating innate freezing behavior to the general threats rather than specific to the fox urine.

## Discussion

Here we show that ACC and its input to the BLA control innate fear response in mice. By using projection-based optogenetic manipulation approaches, we found that inactivation of ACC inputs in the amygdala caused the enhancement of innate fear response. This result provides convincing evidence supporting our conclusion. Given our optogenetic manipulation of ACC terminals in the amygdala included both BLA and CeA areas, it is possible that ACC input to the CeA, although it is very sparse, may also be involved in such control process. It could be that such enhancement of freezing may be a confounding effect resulting from interference of ACC input function unrelated with fear response control. However, this possibility is unlikely because identical optogenetic inactivation of ACC somata alone did not induce freezing in the absence of threat and had no effect on freezing response to a conditioned stimulus. In addition, optogenetic manipulation of the ACC cell somata or ACC-BLA projection did not change general anxiety level or locomotor activity as shown in the open field test.

Considering previously suggested role of ACC for affective pain information processing in fear conditioning[25,41,42], increase rather than decrease of freezing response by the inhibition of ACC-BLA projection was somewhat an unexpected result. The underlying circuit mechanism is unknown in our study. One possible idea is that ACC-BLA projection may control the anxiety provoked in threat situation. In support of this possibility, a previous study[7] showed that activation of the IL-BMA amygdala pathway decreases basal levels of anxiety and also mediates the inhibition of conditioned freezing during extinction, suggesting the possible link between anxiety control and fear response. Likewise, another study showed that activation of the projection terminals of BLA neurons in the CeL (BLA-CeL) has anxiolytic effects[43]. The upstream brain structure controlling anxiolytic BLA-CeL microcircuit activity has not yet been identified. It might be interesting to see whether the BLA neurons mediating such anxiolytic control receive ACC inputs. An alternative possibility is that ACC-BLA circuit directly inhibits the circuitry that

generates freezing behavior. The CeM plays a major role in the expression of freezing[21,28], and a recent study revealed that a circuit pathway from the CeL and CeM to the ventrolateral periaqueductal gray to the pre-motor nucleus of the medulla generates freezing behavior[44]. Considering the known feed-forward inhibitory microcircuit between the CeL and CeM[43,45], it is possible that ACC-BLA circuit activation might inhibit CeM output (through the CeL) to the downstream freezing behavior-generating circuit.

We show that inhibition of the ACC-BLA projection increases innate freezing whereas its activation decreases it. This finding suggests an interesting idea that inhibition of ACC-BLA pathway may be instrumental for innate fear behavior expression and that ACC local inhibitory interneurons are involved during innate fear behavior. Due to the technical limitations, it is unknown in our study whether ACC-BLA projection is inhibited in response to the innate threats. Nevertheless, consistent with the idea, our immunostaining results (activation of GAD67+cells in the ACC to the fox urine) support the possible involvement of local GABAergic interneurons for the innate fear behavior expression. Although the connectivity between the GABAergic interneurons (activated by innate threats) and ACC-BLA cells is unknown in our study, it is possible that such local inhibitory interneurons directly inhibit ACC-BLA cells to promote freezing response. This is an important hypothesis that needs to be addressed in the future study.

A previous study in rat[46] investigated the effect of transient activation or inactivation of the rostal part of ACC (rACC) by using drug on the acquisition of auditory conditioned fear in rats. Their results show that the inactivation of the rACC decreased conditioned freezing measured during training. Conversely, activation increased conditioned freezing response. These data seem to be contradictory with our results. Moreover, a recent study in mice shows a role of ACC-BLA circuit for encoding socially derived aversive cue information in observational learning[47]. One possible explanation is that ACC may have different regulatory roles for the expression of conditioned fear during training and during test 24 h after training. Indeed, the identical inactivation of the rostral ACC 24 h after training does not affect the expression of conditioned freezing[46], which is consistent with our eNpHR3.0 results. An alternative explanation is that the inconsistency may result from the difference in the ACC area manipulated in two studies. We targeted a more caudal part of the ACC, but the study by Bissiere et al. targeted the rostral ACC. In this sense, we propose a hypothesis that the caudal and rostral ACC in rodents may have distinct roles: the caudal ACC controls innate fear response, while the rostral ACC controls the acquisition of conditioned fear perhaps by processing pain information. Notably, another previous study in rat reports that pharmacological activation of the rostral ACC, but not caudal ACC, promotes acquisition of fear

**Fig. 7** Optogenetic manipulation of the ACC-BLA projection changes freezing response to an innate threat. **a** Illustration of eNpHR3.0-based inhibition strategy of the ACC terminals in the amygdala. **b** Representative confocal images showing eNpHR3.0-eYFP terminals in the amygdala. Scale bar, 500 µm. **c** Experimental procedure for fox urine test with photoinhibition of ACC-amygdala projection. **d–f** Percentage of time spent freezing during the fox urine test was selectively increased by photoinhibition in the eNpHR3.0 (ACC projection) group ($n = 10$) (**d**), but not in the eGFP (ACC projection) group ($n = 10$) (**e**) or eNpHR3.0 (PL projection) group ($n = 7$) (**f**). Wilcoxon matched-pair rank tests for eNpHR3.0 ACC projection ($P = 0.0020, = 0.0039, = 0.019$). Paired t-tests for eGFP ACC projection ($P = 0.1736, = 0.4154, = 0.1171$) and eNpHR3.0 PL projection ($P = 0.0024, = 0.0181, = 0.0016$). **g** Light-induced change in percentage of freezing behavior for **d–f**, calculated by subtracting freezing values within the first OFF-ON cycle. One-way ANOVA ($F_{(2, 24)} = 19.65, P < 0.0001$) and Tukey's post hoc test (eNpHR3.0 ACC projection vs. eGFP ACC projection, $P < 0.0001$; eNpHR3.0 ACC projection vs. eNpHR3.0 PL projection, $P < 0.0001$; eGFP ACC-projection vs. eNpHR3.0 PL-projection, $P = 0.7925$). **h** Experimental procedure for fox urine test with ChR2-based photoactivation of ACC terminals in the amygdala. **i, j** Percentage of time spent freezing during the fox urine test was decreased in the ChR2 group ($n = 8$) (**i**) but not in the eGFP group ($n = 7$) (**j**). Paired t-tests for ChR2 group ($P = 0.0007, = 0.0815, = 0.0488$) and eGFP group ($P = 0.5612, = 0.6859, = 0.5114$). **k** Light-induced change in percentage of freezing behavior for **i** and **j**, calculated by subtracting freezing values within the first OFF–ON cycle. Unpaired t-test ($P = 0.0008, t = 4.370, df = 13$). Line graphs are expressed as mean ± s.e.m. Box-whisker plot is expressed as median, interquartile range with 5-95 percentile distribution. *$P < 0.05$, **$P < 0.01$, ***$P < 0.001$

memory[48]. Together, there might be functionally distinct, multiple cell populations in the ACC that mediate different roles in fear processing.

Prefrontal brain structures are thought to mediate top-down control of fear responses in threat situations. In rodents, the IL is well established for its role in the inhibition of learned fear during extinction, likely through a major output to the BMA[6,7,49]. It is still unclear how the ACC-BLA projection is functionally different from the IL-amygdala projection in the control of fear response. It is possible that the two circuits may have separate roles for the regulation of innate (ACC) versus learned (IL) fear response. Alternatively, the IL may be specifically recruited for

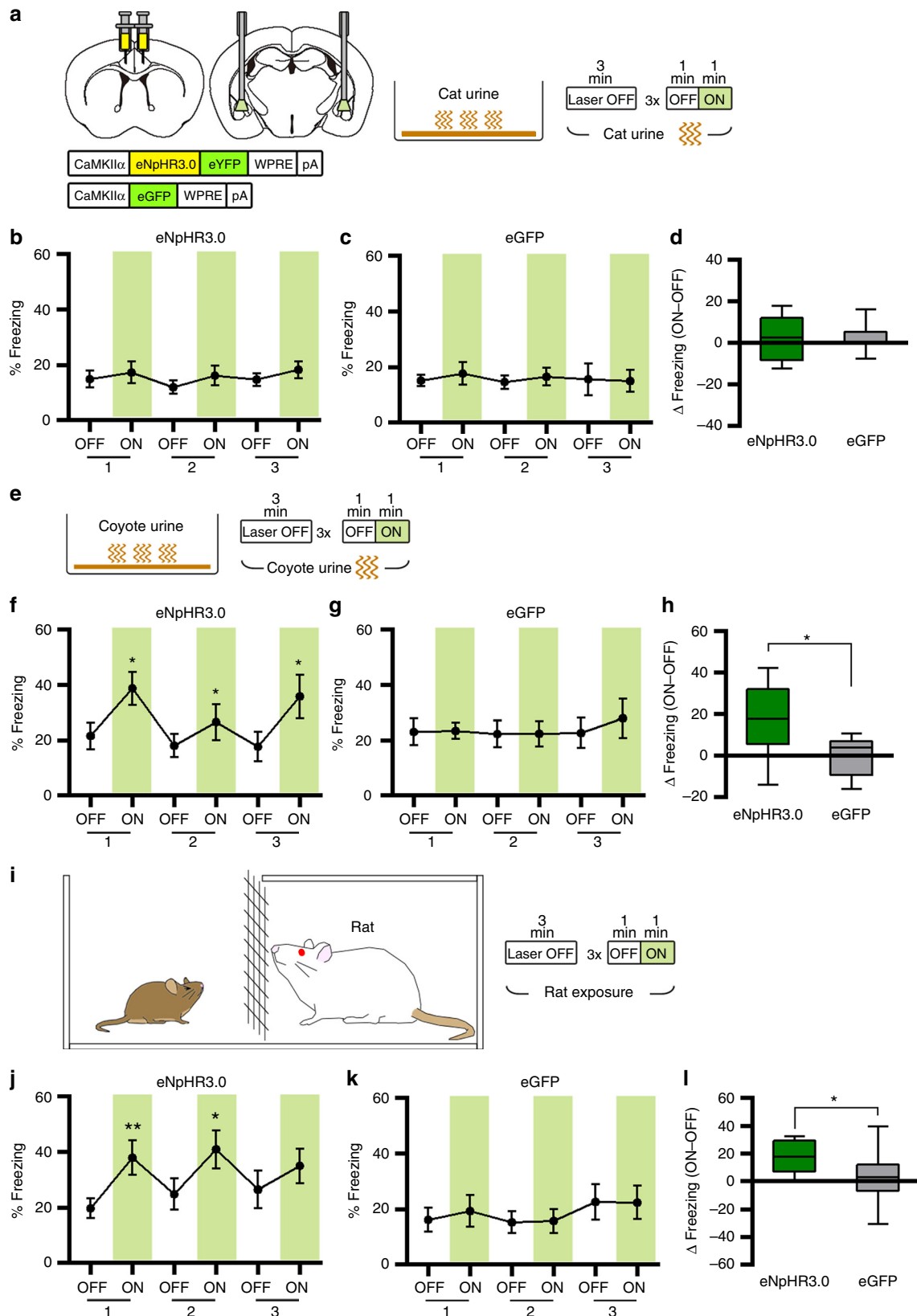

the control of fear expression only after extinction[49]. Because the activation of ACC-BLA circuit led to a robust inhibition of conditioned freezing, it is possible that ACC-BLA circuit may be involved in the reduction of conditioned freezing during extinction training. Although we focused on the BLA as a downstream brain area of ACC, the ACC also sends its projection to various target areas in the neocortex and brainstem. Previous data based on the whole-brain analysis of Synaptophysin-GFP tracing show that some of fear-related target regions, including the PL, IL, PAG and dorsomedial thalamus, display even stronger intensities than the BLA[50]. Thus, it is possible that these ACC projections may also be involved in the control of innate fear behaviors.

Here our findings establish that a projection input from the ACC to the BLA is a critical component of the brain circuitry controlling innate fear responses to predator-related olfactory stimulus in mice, which might be relevant to fear- and anxiety-related psychiatric disorders in humans.

## Methods

**Mice**. $129 \times C57Bl/6$ mice (male or female, 2–3 months of age) were group-housed on a 12-h light-dark cycle under constant temperature ($21 \pm 1$ °C) and humidity (40–60%). Food and water were supplied ad libitum throughout the experiments. All procedures for animal experiments were approved by the KAIST Institutional Animal Care and Use Committee. For ex vivo patch-clamp experiment using high-titer AAV1-hSyn-Cre, a transgenic mouse line (Ai9) containing floxed-stop-tdTomato (B6;129S6-$Gt(ROSA)26Sor^{tm9(CAG-tdTomato)Hze}$/J; The Jackson Laboratory) was used. Mice were randomly allocated to the experimental versus control groups, except for that number of male and female mice was nearly matched to 1:1 ratio for behavior experiments.

**DNA constructs**. The original eGFP-Syb2-expressing plasmid (pAAV-hSyn-mCherry-IRES-eGFP-Syb2) was provided by Daesoo Kim (KAIST), and the hSyn promoter was replaced by CaMKIIα promoter by sub-cloning. pAAV-EF1a-DIO-hChR2(H134R)-EYFP was kindly provided by Karl Deisseroth (Stanford University).

**Viruses**. For customized production of AAV1, pAAV and AAV1 serotype packaging vectors were co-transfected into human embryonic kidney cells by calcium phosphate transfection. After 60–72 h of incubation, AAV was purified from transfected cells by iodixanol-gradient ultracentrifugation. Purified solution was dissolved in phosphate-buffered saline (PBS), and viral titer was measured by quantitative PCR (Qiagen, Rotor-gene Q). AAV solution with a titer range of 3 to $8 \times 10^{11}$ viral genome per ml was used for the expression of genetic materials. Solution with excessive titer was re-dissolved in PBS by 2- to 3-fold. Prepared AAV materials were aliquoted and stored at −80 °C until use.

High titer AAV1-hSyn-Cre ($3.2 \times 10^{13}$ GC/ml) and AAV2-EF1α-ChR2-eYFP ($3.6 \times 10^{12}$ GC/ml) solutions were purchased from Vector BioLab and UNC Vector Core, respectively. CAV2-Cre, with a titer of $6 \times 10^{12}$ viral genome per ml, was purchased from Montpellier vector platform (PVM). Virus solution was aliquoted and stored at −80 °C until use.

**Behavior**. Contextual fear conditioning was performed in a rectangular acrylic chamber ($345 \times 90 \times 160$ mm, W × D × H, custom-made) equipped with two sets of metallic grid floors (Coulbourn). On Day 1, mice were individually placed into the

box and received three foot shocks (0.5 mA, 2 s) at 30-s intervals. The initial shock onset was 2 min after box entry. Day 2, The contextual fear retention test was performed in the same chamber. Three minutes after entry, freezing responses were monitored over three light delivery cycles (1 min OFF–1 min ON).

Auditory fear conditioning was performed in a square-shaped metallic chamber ($178 \times 178 \times 305$ mm, W × D × H, Coulbourn) equipped with a grid floor (Coulbourn). Day 1, as a background odor, 70% ethanol was poured into a tray below the grid floor. Two minutes after chamber entry, a pure-tone stimulus (2.1 kHz, 30 s, 70 dB) was presented. A foot shock (0.5 mA, 2 s) was delivered during the last 2 s of the tone delivery (co-termination). Day 2, The auditory fear retention test was performed in a rectangular acrylic chamber ($345 \times 90 \times 160$ mm, W × D × H, custom-made). To provide a different context, the floor was substituted with a white glossy acrylic plate, and 70% ethanol was not presented. Three minutes after entry, freezing responses to the tone stimulus (2.1 kHz, 6 min, 70 dB) were monitored over three light delivery cycles (1 min OFF–1 min ON).

The fox urine exposure test was performed in a similar manner as previously described[27]. Three sheets of KimWipes laboratory tissue were fully soaked in 10 ml of 100% red fox urine (Leg Up Enterprise) and evenly piled over the floor of a plastic cage (1-L-sized regular mouse cage, ZoonLab), and mice were placed in the cage for behavioral testing. In control experiments with water, tissue sheets were soaked in distilled water instead of fox urine. Exposure to other predator urine was performed by the same protocol, except for the use of cat (10 ml of 100% wild cat urine, PeteRickards) or coyote urine (10 ml of 100% coyote urine, Leg Up Enterprise). For optogenetic experiments, mice were acclimated for 3 min after entry, and freezing responses were monitored over three light delivery cycles (1 min OFF–1 min ON).

The fox urine avoidance test was performed in a similar procedure as described in previous literature[29,39], but the size of the apparatus was modified for mice. A square open field arena (white non-glossy acrylic box, $450 \times 450 \times 250$ mm, W × D × H, custom-made) was used as the testing chamber. At the up-left or down-right side of the corner, a 35-mm diameter petri dish which was filled with 1 ml of 100% red fox urine (Leg Up Enterprise) or distilled water (control) was placed. The placement of sample dish was counterbalanced for every trial. Mice were introduced to the opposite corner and monitored for 3 min (or total 6 min for optogenetic sessions; 3 min OFF and 3 min ON). The testing arena was cleansed with 10% ethanol after every trial.

Similar to a previous literature[51], the rat exposure test was performed in a two-chambered acrylic cage ($250 \times 450 \times 250$ mm, W × D × H, custom-made) which was partitioned by a metallic wire mesh. As a modification, hiding tunnel and shelter (for mice) were excluded. A mouse was introduced to one partition of the cage, and then a live rat (Sprague-Dawley, 10-week, male, not treated with amphetamine) was introduced to the opposite partition behind the wire mesh. For optogenetic experiments, mice were acclimated for 3 min, and freezing responses were monitored over three light delivery cycles (1 min OFF–1 min ON). The testing chamber was cleansed with 10% ethanol after every trial. A new rat was used after every three trials.

Open field test was performed in a square arena (white non-glossy acrylic box, $450 \times 450 \times 250$ mm, W × D × H, custom-made). Mice were introduced to the apparatus, and monitored over three light delivery cycles (3 min OFF–3 min ON). The testing arena was cleansed with 10% ethanol after every trial.

For behavior experiments with optogenetic stimulation, laser light was generated using CrystaLaser diodes (CL561-050-O for 561 nm green light, CL473-050-O for 473 nm blue light) and delivered through surgically implanted fiber-optic elements (Doric lenses). For photoactivation of ChR2-expressing somata and terminals, blue light (473 nm wavelength; 0.2 mW at fiber tip for somata or 0.5 mW at fiber tip for terminals) was delivered as pulses (10 ms width, 10 Hz). For photoinhibition of eNpHR3.0-expressing somata and terminals, green light (561 nm wavelength; 5 mW at fiber tip) was continuously illuminated.

Mouse behavior was recorded by a CCD camera from above at four frames per second. For scoring of freezing behavior, video files were randomized before

**Fig. 8** Photoinhibition of the ACC-BLA projection induces increased freezing response to coyote urine and live rat, but not to cat urine. **a** Cat urine exposure test with photoinhibition of the ACC-BLA projection. **b, c** Percentage of time spent freezing during the cat urine test was not changed in both the eNpHR3.0 group ($n = 6$) (**b**) and eGFP group ($n = 7$) (**c**). Paired $t$-tests for eNpHR3.0 group ($P = 0.6343, = 0.2623, = 0.2775$) and Wilcoxon matched-pairs rank tests for eGFP group ($P = 0.6875, = 0.6875, > 0.9999$). **d** Light-induced change in percentage of freezing behavior for **b** and **c**, calculated by subtracting freezing values within the first OFF–ON cycle. Unpaired $t$-test, ($P = 0.9761, t = 0.0306$, df = 11). **e** Coyote urine exposure test with photoinhibition of the ACC-BLA projection. **f, g** Percentage of time spent freezing during the coyote urine test was specifically increased in the eNpHR3.0 group ($n = 9$) (**f**), but not in the eGFP group ($n = 7$) (**g**). Paired $t$-tests for eNpHR3.0 group ($P = 0.0173, = 0.0470, = 0.0217$) and eGFP group ($P = 0.9205, = 0.9281, = 0.5379$). **h** Light-induced change in percentage of freezing behavior for **f** and **g**, calculated by subtracting freezing values within the first OFF–ON cycle. Unpaired $t$-test ($P = 0.0368, t = 2.307$, df = 14). **i** Rat exposure test with photoinhibition of the ACC-BLA projection. **j, k** Percentage of time spent freezing during the rat exposure test was specifically increased in the eNpHR3.0 group ($n = 10$) (**j**), but not in the eGFP group ($n = 10$) (**k**). Wilcoxon matched-pairs rank tests for eNpHR3.0 group ($P = 0.0039, = 0.0488, = 0.2324$) and eGFP group ($P = 0.4316, > 0.9999, > 0.9999$). **l** Light-induced change in percentage of freezing behavior for **j** and **k**, calculated by subtracting freezing values within the first OFF–ON cycle. Unpaired $t$-test ($P = 0.0356, t = 2.272$, df = 18). Line graphs are expressed as mean ± s.e.m. Box-whisker plot is expressed as median, interquartile range with 5-95 percentile distribution. *$P < 0.05$, **$P < 0.01$

counting, and duration of freezing was manually scored using a computer-based stopwatch software. For tracking of mice during avoidance test and open field test, video files were automatically analyzed using EthoVision software (Noldus).

**Surgery for behavior and tracing experiments.** For optogenetic manipulation experiments, mice were anaesthetized by intraperitoneal injection of sodium pentobarbital (83.3 mg/kg) and fixed in a stereotaxic surgery frame. For ACC infection, 1.5 μl AAV solution was injected at the following sites relative to bregma: anterior-posterior (A/P)+0.8 mm, medial-lateral (M/L) ±0.35 mm, and dorsal-ventral (D/V) −1.8 mm. We used the viral titer and coordinate conditions for optimized caudal-specific infection of the ACC (covering A/P+1.4 to+0.0 mm). In a control experiment targeting the PL area, 1.0 μl AAV solution was injected at A/P +1.7 mm, M/L ±0.35 mm, and D/V −2.0 mm. Two weeks after viral injection surgery, mice underwent surgery again for implanting fiber-optic elements (Doric lenses, 200 μm core diameter, 0.22 NA) into ACC or amygdala coordinates (ACC: A/P+0.8 mm, M/L ±0.35 mm, D/V −1.5 mm; amygdala: A/P −1.6 mm, M/L ±3.35 mm, D/V −4.5 mm). Mice recovered in their home cages for 1 week before behavioral experiments.

For eGFP-Syb2 expression, mice were injected with 1.5 μl AAV-CaMKIIα-mCherry-IRES-eGFP-Syb2 at bilateral ACC coordinates and allowed to recover in their home cages for 6–8 weeks until sacrifice.

For CAV2-Cre-dependent expression, mice were injected with 0.3 μl CAV2-Cre into the bilateral BLA (A/P −1.6 mm, M/L ±3.35 mm, D/V −4.7 mm) and 1.5 μl AAV-EF1α-DIO-ChR2-eYFP (or AAV-EF1α-DIO-ChR2-eYFP) into the bilateral ACC (A/P+0.8 mm, M/L ±0.35 mm, D/V −1.8 mm). Two weeks later, mice for behavioral experiments underwent surgery for implanting fiber-optic elements into the ACC (A/P+0.8 mm, M/L ±0.35 mm, D/V −1.5 mm) and were allowed to recover in their home cages until behavioral experiments. Mice for retrograde tracing experiments were sacrificed 2 weeks after the injection surgery without undergoing fiber implantation surgery.

**Surgery for ex vivo patch-clamp recording experiments.** For experiments using AAV-CaMKIIα-ChR2, wildtype mice of 129 × C57Bl/6 background were used. Mice of 4-week age were injected with 1.5 μl solution of AAV1-CaMKIIα-ChR2 into the bilateral ACC (modified coordinate for juvenile mice: A/P+0.7 mm, M/L ±0.3 mm, D/V −1.7 mm). Mice were recovered in their home cages for 4 weeks until the slice recording experiments.

For experiments with anterograde AAV vector, both high titer AAV1-hSyn-Cre (100 nl) and AAV2-EF1a-DIO-ChR2-eYFP (50 nl) were pressure-injected into the bilateral ACC (A/P+0.7 mm, M/L ±0.3 mm, D/V −1.7 mm) of male Ai9 mice (5 −6 weeks old), with a thin glass pipette using Nanojector III (Drummond) and the stereotaxic instrument (Kopf). At least 4 weeks were allowed for anterograde transfer of AAV1-hSyn-Cre and full expression of ChR2-eYFP in the ACC and tdTomato in the ACC-connected postsynaptic neurons.

**Histology.** Mice were transcardially perfused with 4% paraformaldehyde (PFA) dissolved in PBS. Dissected brains were incubated in 4% PFA/PBS for an additional 8 h after perfusion. For immunohistochemistry, brains were dehydrated in 30% sucrose/PBS for 48 h and sectioned coronally (40 μm thickness) on a −20 °C cryostat. Brains used for optogenetic experiments were sectioned using a Vibratome (Leica) without the dehydration step. Prepared brain sections were stored in 0.02% sodium azide dissolved in PBS. For histological analysis, sections were cover-slipped with Vectashield mounting solution (Vector Laboratories) and observed under a Nikon Eclipse 80i fluorescence microscope or Carl Zeiss LSM780 confocal microscope. Through post hoc analysis of the section histology, two mice from Fig. 6d and one mouse from Fig. 6g were excluded from the behavior data due to AAV expression in the rostral ACC and PL area.

**Immunohistochemistry.** Mice were exposed to fox urine or water for 1 h, and transcardially perfused with 4% PFA/PBS. Brain sections were prepared as described in Histology section of Methods above. Three or four sections per mouse (covering the PAG, A/P −4.2 mm to −4.8 mm from bregma) were used for staining experiment. For single staining for c-Fos, sections were incubated with primary antibody against c-Fos (rabbit, 1:1000, catalog # sc-52, Santa Cruz; diluted in blocking solution containing 2% goat serum, 0.1% bovine serum albumin, and 0.3% triton X-100; 16 h), and then with secondary antibody against rabbit IgG (goat, 1:500, conjugated with Alexa Fluor 594, catalog # A11037, Molecular Probes; 2 h). Between antibody incubation steps, samples were rinsed in PBS solution 3–4 times. From the immunostained sections, tile-scan images for covering the dorsal PAG and ventral PAG were acquired by confocal observation at 10× magnification (LSM780, Carl Zeiss). To calculate the c-Fos density, the number of c-Fos positive nuclei was automatically quantified using Nikon elements BR software.

For double staining for c-Fos and GAD67, three sections per mouse (covering the caudal ACC, A/P+1.10 mm to+0.8 mm from bregma) were used for staining experiment. Sections were incubated with primary antibodies (rabbit anti-c-Fos, 1:1000, catalog # sc-52, Santa Cruz; and mouse anti-GAD67, 1:1000, catalog #MAB5406, Millipore; 16 h), and then with secondary antibodies (goat anti-rabbit IgG, 1:500, conjugated with Alexa Fluor 488, catalog # A11008; goat anti-mouse IgG, 1:500, conjugated with Alexa fluor 594, catalog # A11005; 2 h). From the

immunostained sections, tile-scan images for covering the left and right ACC fields were acquired by confocal observation at 40× magnification (LSM780, Carl Zeiss). Location of GAD67+cell was manually identified (at Alexa 594 channel) following constant criteria (>5 μm nucleus diameter, >2 μm cell body thickness). Next, the number of c-Fos+nuclei surrounded by the GAD67+cell bodies was quantified, and divided by the total count of GAD67+cells per mouse. For double-blinded analysis, the counter was not informed with the experimental hypotheses and group assignment.

For double staining for VGLUT1 and VGAT, four sections per mouse (A/P −1.20 mm to −1.60 mm from bregma) were used for staining experiment. Sections were incubated with primary antibodies (rabbit anti-VGLUT1, 1:1000, catalog #135 303; and mouse anti-VGAT, 1:500, catalog #131 011; Synaptic Systems) for 16 h at room temperature. Sections were then incubated with secondary antibodies (Alexa Fluor 405-conjugated anti-rabbit IgG, 1:1000; and Alexa Fluor 647-conjugated anti-mouse IgG, 1:500; Molecular Probes) for 2 h at room temperature. From the bilateral BLA area of the unstained sections, serial Z-stack images (1 μm thickness, 6-9 planes for each trial) were acquired by confocal observation at 100× magnification (LSM780, Carl Zeiss). In the figure, signals are presented in pseudo-colors for better visibility (eGFP-Syb2 in yellow; VGLUT1-Alexa Fluor 405 in magenta; VGAT-Alexa Fluor 647 in cyan).

**Ex vivo electrophysiology.** All chemicals were purchased from Sigma unless indicated. The standard artificial cerebral spinal fluid (ACSF) consisted of (in mM) 124 NaCl, 2.5 KCl, 1.2 NaH$_2$PO$_4$, 24 NaHCO$_3$, 5 HEPES, 2 CaCl$_2$, 2 MgCl$_2$, and 13 glucose (pH 7.3). Mice were deeply anesthetized with isoflurane and transcardially perfused with ~20 ml of the slicing ACSF containing (in mM) 93 N-methyl-D-glucamine (NMDG)-Cl, 2.5 KCl, 1.2 NaH$_2$PO$_4$, 30 NaHCO$_3$, 20 HEPES, 5 sodium ascorbate, 2 Thiourea, 3 sodium pyruvate, 12 N-acetyl-L-cysteine (NAc), 0.5 CaCl$_2$, 10 MgCl$_2$, and 25 glucose (pH 7.3) before the brain was dissected. Coronal slices containing the amygdala (400-μm thick) were prepared using a VF-200-OZ Compresstome (Precisionary) using the slicing ACSF and recovered at 30–32 °C in the recovery ACSF (in mM; 104 NaCl, 2.5 KCl, 1.2 NaH$_2$PO$_4$, 24 NaHCO$_3$, 5 HEPES, 5 sodium ascorbate, 2 Thiourea, 3 sodium pyruvate, 2 CaCl$_2$, 2 MgCl$_2$, and 13 glucose; pH 7.3) for 1 h before recording. Slices were then placed in a recording chamber, submerged, and continuously perfused (2–3 ml/min) with the oxygenated standard ACSF (20–25 °C). To isolate glutamatergic synaptic responses, 10 μM bicuculline (Tocris; GABA-A receptor antagonist) was added to the standard ACSF. To test whether light-evoked synaptic responses were mediated by glutamatergic synaptic transmission, slices were treated with 2,3-dihydroxy-6-nitro-7-sulfamoyl-benzo[f]quinoxaline-2,3-dione (NBQX, 10 μM; Tocris; AMPA receptor antagonist) and D-2-Amino-5-Phosphonovaleric acid (D-AP5, 50 μM; Tocris; NMDA receptor antagonist). To isolate monosynaptic EPSP responses elicited by light stimulation, both tetrodotoxin (TTX; 1 μM, Alomone Labs; voltage-gated sodium channel blocker) and 4-aminopyridine (4-AP; 100 μM; voltage-gated potassium channel blocker) were included.

Whole-cell current-clamp recordings were made with a Multiclamp 700B amplifier (Molecular Devices). Data were filtered at 2 kHz, digitized at 1–5 kHz, stored on a computer, and analyzed offline using pCLAMP 10 (Axon Instruments). BLA neurons were visualized with infrared differential interference optics equipped with a 40× water-immersion objective lens. Borosilicate glass patch electrodes had a resistance of 3–5 MΩ after filling with pipette solution containing (in mM) 140 K-gluconate, 5 KCl, 0.2 EGTA, 2 MgCl$_2$, 4 Mg-ATP, 0.3 Na$_2$-GTP, 10 Na$_2$-phosphocreatine, and 10 HEPES (pH 7.3, 290-300 mOsm).

To induce light-evoked PSPs, high-power blue LED light (at 470 nm; X-Cite) was delivered to the slice through the standard optic path in the microscope (Nikon). Using a 40× objective lens, this configuration could deliver blue light at ~8 mW/mm$^2$ over a ~0.20 mm$^2$ area of the recording site. With 0.2–1 ms durations of LED illumination, these conditions were sufficient to elicit stable excitatory PSPs ranging from 1.0 to 10 mV.

**Statistical analysis.** Statistical analyses were performed using GraphPad Prism 6. Normal distribution of each dataset was tested by Shapiro-Wilk normality test ($\alpha = 0.05$). Where the normality test failed ($P < 0.05$), appropriate non-parametric tests were used. To investigate the light-induced effect, freezing scores within each OFF–ON cycle was compared in a pairwise manner (paired $t$-test or Wilcoxon matched-pair rank test). For group-wise comparison purpose, differential freezing score ($\Delta$) was calculated by subtraction between the first OFF-epoch and ON-epoch scores. These scores were compared by two-tailed unpaired $t$-test or Mann–Whitney U test. Results in avoidance test and open field test were compared between experimental (eNpHR3.0 or ChR2) and eGFP control groups using repeated measures (stacked) two-way ANOVA with Sidak's post hoc test. Sample size was not predetermined by statistical methods.

**Data availability.** The data that support the findings of this study are available from authors upon reasonable requests.

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

## Acknowledgements

The authors thank all laboratory members for critical discussions and helpful comments. The authors also thank M.J. Kim and H.-Y. Cho for assistance with the preparation of AAV materials. This work was supported by grants from the National Research Foundation of Korea (NRF) funded by the Korean Ministry of Science, ICT (MSI; 2014R1A2A1A10053821), the Brain Research Program through the NRF (2017M3C7A1031322), an Intelligent Synthetic Biology Center of the Global Frontier Project funded by the MSI (2011-0031955), the KBRI basic research program through Korea Brain Research Institute funded by the MSI (18-BR-01-03 and 18-BR-04-01), and the KAIST Future Systems Healthcare Project. J.J. was supported by a National Junior Research Fellowship (2013H1A8A1003842) through the NRF.

## Author contributions

J.-H.H. designed and directed the study. J.J. performed surgical, behavioral, optogenetic and tracing experiments. H.L. and H.P. performed electrophysiological experiments and tdTomato tracing experiments. M.S.K. assisted with DNA sub-cloning and preparation of

AAV materials. H.-S.L. performed c-Fos counting analysis. J.J., H.P., and J.-H.H. analyzed the data and wrote the manuscript.

## Additional information

**Competing interests:** The authors declare no competing interests.

