## [Peer Review File · Nature Communications]

Reviewers' comments:

Reviewer #1 (Remarks to the Author):

In this manuscript Jhang et al. identify a projection from anterior cingulate cortex (ACC) to basolateral amygdala (BLA) that controls freezing to a predator odor (fox urine). The authors used optogenetic manipulations to show that inhibition of either the ACC or the ACC-BLA projection enhances freezing to fox urine, but not to conditioned fear stimuli. They also show that excitation of either the ACC or the ACC-BLA projection decreases freezing to both conditioned stimuli and fox urine. Using antibody staining and patch clamping the authors argue that this projection is glutamatergic.

Overall this is a straight-forward paper that would be of interest to the readership of the journal. This work identifies a novel pathway that controls defensive reactions to predators. Avoiding predators is a key ability for promoting survival that has not been extensively studied using modern standard techniques of systems neuroscience such as optogenetics.

There are, however, a few additional experiments that need to be done to strengthen the authors' claims and demonstrate the specificity of the manipulations.

1. Show if activation of the ACC-BLA projection:

1a. Inhibits freezing to a different predator odor (for example, cat urine)

1b. Alters anxiety-behaviors in the elevated plus maze or open field

1c. Inhibits other defensive behaviors in the presence of predator cues, such as escape, avoidance of fox urine or risk-assessment behaviors.

1d. Inhibits reactions during exposure to a live predator (a rat, for example).

2. Investigate the specificity of this manipulation by testing if prelimbic cortex input to the BLA changes freezing to fox urine.

3. The patch-clamping characterization of the ACC-BLA projection in Fig. 4i should be expanded.

3a. The authors claim that "To confirm the direct glutamatergic nature of ACC-BLA synaptic connections, we performed slice patch-clamp recording of BLA neurons". However their measures do not demonstrate monosynaptic connectivity. Repeat figure 4i in the presence of tetrodotoxin and 4-aminopyridine to show that the BLA light-induced response is monosynaptic.

3b. Show BLA cell responses to ACC terminal optical stimulation with the same parameters used in the behavioral experiments.

4. The authors should add a sentence or two discussing what, in their view, is the functional difference between the projections of the IL and ACC to the amygdala.

Minor points.

1. Please provide the titer of CAV injections

2. The authors should remove the text " Thus, our circuit analyses indicate that the BLA is the major ACC projection target in the amygdala and receives direct glutamatergic excitatory input". Their data do not support this claim, as the authors did not demonstrate that the BLA received stronger ACC input than other amygdala nuclei. To support this claim the authors would have to systematically compare the strength of ACC input to various amygdala nuclei.

Other notes.

The statistical analysis used was appropriate.

Reviewer #2 (Remarks to the Author):

In this study Jhang et al. analyze the role of ACC and ACC-BLA projection neurons in innate and conditioned fear responses. The study is presenting strong and interesting results obtained with optogenetic manipulation of the ACC. The author found that inhibition and activation of the glutamatergic neurons of the ACC, respectively increase and decrease threat response (freezing) to an innately threatening stimulus (fox urine). They also found that only optogenetic activation of these neurons had an impact of conditioned fear responses, while optogenetic inhibition did not alter the retrieval of the conditioned fear. However, the projection-specific study, including the anatomical and electrophysiological part are not carefully performed. The optogenetic inhibition of ACC-BLA needs to be re-done with appropriate placement of the optic fiber and additional analysis and moderation of the claims are needed for the anatomy and electrophysiology in order for the study to be suitable for publication in nature communication.

_____Major comments_____

The authors used an elegant viral strategy to presynaptically visualize ACC neurons the axonal projection in the amygdala, the authors claim the ACC is mainly projecting to the BLA and not to the central amygdala (CeL, CeC and CeM). They show a good example of this in Figure 4b, but there is no quantification across the different subfields of the amygdala (fluorescence / mm²) and even more importantly in different mice to identify how reproducible this observation is (see figure 3 and S3 of Beyeler et al., 2016). My concerns are even strengthened by the fluorescence image in Figure 4h where you can see almost more fluorescence in the central amygdala than in the BLA.

The ACC also projects to many other downstream regions involved in fear-related behaviors such as, the mPFC or the ventral periaqueductal grey or the hippocampus. The eGFP-Syb2 fluorescence should also be quantified in these different regions and compared to the BLA.

Regarding the projection specific experiments, the authors injected 1000 nL of CAV2-cre virus in the BLA, which is a huge volume for an injection in the mouse, and questions the specificity of the staining. Indeed, 300 nL (less than 1/3) is the amount used in many studies in the field performing dual virus targeting of projection neurons. Again, the results of the anatomy of ACC-BLA projection experiments remain mainly qualitative. The author mention most of the projectors are ipsilateral, but do not provide counts of cell bodies in the

ipsilateral vs contralateral part, and how this is consistent across animals.

In the electrophysiological recordings the number of neurons is rather low (n=6) and the claims are not supported by the data. Indeed, in the abstract the author say "Circuit tracing and electrophysiological recordings identified a monosynaptic glutamatergic input from a subpopulation of ACC neurons to the BLA". However, to demonstrate that a connection is monosynaptic, the author need to block polysynaptic activation in the slice, which is generally achieved by application of the sodium channel blocker TTX (see Figure 1 of Petreanu et al., 2009). The authors should also quantify the latency of onset of the photocurrent, which could be an indication of mono or polysynaptic transmission.

For the optogenetic inhibition of ACC terminals in the BLA, despite the authors mentioned the strongest inputs in the amygdala is in the BLA, they place the optic fibers in the central amygdala (Supplementary Figure 5). Similarly, to photoactivate the BLA-ACC neurons, the author placed the optic fibers in the central amygdala (Supplementary Figure 6). This compromises the interpretation of all the results of figure 6 and 7.

It is unclear whether ACC inhibition during conditioned and innate fear stimuli was performed in the same experimental group. From the number of mice in the control group (7 for CS and 8 for US) it seems that the mice are from different experimental groups. Because, optogenetic manipulation are depending on many variables (location of viral injection, viral expression, fiber placement..) it would have been a good control to have the manipulations for both the CS and US in the same animals. If this is not the case, please make it clear.

The statistical comparison is not well described and seems not appropriate. Indeed, the authors use multiple paired t-tests for repeated measure of freezing across CS presentations (Figure 1d-e, j-k) where an ANOVA would be more suitable. Moreover, the same data is used in the differential freezing score, therefore, the p values should be corrected for multiple comparison.

_____Minor comments_____

The y-axis for the freezing to the conditioned cue is at 100% (Figure 1d-e) whereas the y-axis is at 60 % for the rest of the figure. This is misleading and suggest the authors want to reduce the perception of the impact of the optogenetic inhibition on the behavioral response to the conditioned cues.

For optogenetic activation the author used blue light (473 nm wavelength, 0.2 mW at fiber tip). The light power seems very low in comparison with previous publications.

This sentence in the introduction is too long: Previous studies in rodents extensively investigated the role of prelimbic (PL) and infralimbic (IL) regions of the mPFC, which are homologous to the ventral-medial prefrontal cortex (vmPFC) and ventral anterior cingulate cortex (vACC) in humans^{1,3}, and their projections in the regulation of fear behavior in

response to learned threat.

Figure 4c: BLA: CAV2-cre should be above the BLA diagram/needle

The titer of the CAV2-cre virus should be specified.

Reviewer #3 (Remarks to the Author):

This manuscript by Jhang et al, investigated the role of the caudal anterior cingulate cortex (ACC) to basolateral amygdala (BLA) projection in innate fear behavior using a combination of anatomical tracing, optogenetic and behavioral approaches. The authors first observed that inhibiting the ACC using optogenetic lead to a surprising enhancement of freezing responses to fox urine and had no effect on cued fear conditioning. The opposite manipulation decreased both innate and conditioned freezing behavior. Using elegant neuroanatomical tracing strategies, they demonstrated that ACC terminals contacted preferentially excitatory BLA neurons and provide excitatory inputs onto these neurons. Finally specific manipulations of the ACC-BLA pathway reproduced the finding observed using manipulation of the entire ACC. Another experiment out of scope of the present study suggested that activation of BLA to ACC inputs also reduced innate fear behavior to fox urine.

The manuscript is potentially interesting but raises several unanswered questions. I have several conceptual and technical remarks.

1) First of all, the main finding of the study is quite surprising as the inhibition of the ACC-BLA pathway increases innate fear behavior whereas its activation decreases it. This suggests that inhibition of the ACC-BLA pathway is instrumental for innate fear behavior expression and that ACC local inhibitory interneurons are involved (activated) during innate fear behavior. It also suggests that under basal conditions the ACC-BLA pathway is only partially inhibited which lead to some innate fear behavior. I think this aspect deserve additional experimental work and will strengthen the impact of the manuscript. The authors could for instance perform some interneuronal recordings/imaging experiments during behavioral training (photometry, calcium imaging, single unit recordings).

2) Upon light activation of the ACC alone or the ACC-BLA pathway, the authors observed a reduction in innate fear behavior. However, this drop in freezing behavior could also be related to non specific motor effects, which are not controlled in the present study.

3) It is largely assumed that fox odor induces an aversive/fear state, yet to my knowledge there is no studies demonstrating that the so called freezing observed upon fox odor exposure correspond really to freezing behavior and not just immobility. These two options could be disentangle using EMG recordings like in Steenland and Zhuo (2009).

4) The last section of the paper is out of focus of the present research and should be

removed from the main manuscript. It raises additional questions and the rational for performing this experiment is absolutely not clear.

5) The results of the present manuscript are quite contradictory with the work from Bissiere et al., *Biol Psy*, 2008, which partially target the caudal ACC. This work should be discussed.

Reviewer #1

1. Show if activation of the ACC-BLA projection:

1a. Inhibits freezing to a different predator odor (for example, cat urine)

We agree with the reviewer that it is important to determine the specificity of fear response control by ACC-BLA projection. To address this concern, as the reviewer suggested, we first investigated whether optogenetic inactivation of the ACC-BLA projection also produces the increase of freezing response to other predator odors. The eNpHR3.0 was expressed in the bilateral ACC, and ACC-BLA projection was inhibited by the 560-nm light illumination in the BLA as before. We first tested the freezing response to the cat urine as the reviewer suggested. In the presence of cat urine [10 ml of 100% wild cat urine (PeteRickards)], mice exhibited no significant change in freezing response with photoinhibition (Figure 1 below). According to previous reports, it is likely that cat urine may induce no or very weak fear response including freezing in rodents (Blanchard et al., 2003; Fendt, 2006). A study by Blanchard et al. shows that cat urine produces no significant freezing response in rats. Moreover, a study by Fendt et al. shows that laboratory-raised rats display innate avoidance response and risk assessing behavior to various predator urines (bobcat, cougar, fox, coyote, and wolf), but not to cat urine.

Figure 1. Photoinhibition of the ACC-BLA projection does not induce increased freezing response under cat urine exposure.

So, to further confirm the specificity, we next adopted coyote urine [10 ml of 100% coyote urine (Leg Up Enterprise)] as another predator odor. Different from cat urine, in this condition, mice displayed significant increase of freezing response with photoinhibition of ACC-BLA projection compared to eGFP control group (Figure 2). These results thus further support that ACC-BLA projection has a general control function for the freezing response to predator odors, but not specific to the fox urine. We included these important data in the Fig. 7 of the revised manuscript.

Figure 2. Photoinhibition of the ACC-BLA projection induces increased freezing response under exposure to coyote urine.

1b. Alters anxiety-behaviors in the elevated plus maze or open field

As the reviewer suggested, we performed open field behavior test to investigate whether

optogenetic inhibition or activation of the ACC-BLA projection alters the anxiety-related behavior (center time). Neither photoinhibition (in eNpHR3.0 group, Figure 3c below) nor -activation (in Chr2 group, Figure 3f) of the ACC-BLA projection changed time spent in center area compared to the eGFP control group. Moreover, we found that locomotor activity is also unchanged by ACC-BLA manipulation. These results support that that activation of the ACC-BLA projection does not alter general anxiety or locomotor behavior and that the effect of ACC-BLA manipulation on threat-induced freezing response was specific. We added these new control data in the Fig. 6 of the revised manuscript.

1c. Inhibits other defensive behaviors in the presence of predator cues, such as escape, avoidance of fox urine or risk-assessment behaviors.

To address this question raised by the reviewer, we performed avoidance behavioral test where we examined the effect of photoinhibition of ACC-BLA projection on the avoidance response of mice to the fox urine. For this experiment, we used a method described in the previous studies in rats (Fendt, 2006; Wernecke et al., 2015) with slight modifications (please see the methods section for the details in the revised manuscript). In brief, we placed a petri-dish containing either fox urine or water as a control at one corner (sample quadrant) and measured the time duration mice spent at that sample quadrant. We first validated our behavior set-up with wild type mice. Consistent with the previous report, we also observed a similar avoidance behavior to the fox urine in wild type mice compared to the water control. Next we performed same behavioral test with mice expressing eNpHR3.0 or eGFP as a control in ACC-BLA projection. We found no significant difference in the avoidance behavior between eNpHR3.0 and eGFP control group in both light off and on sessions. These results suggest that the control function of ACC-BLA may be specific to the freezing behavior, but not general to various fear responses. Nevertheless, our results do not rule out the possibility that ACC-BLA projection may be involved in the control of other defensive behaviors such as risk-assessment behavior. These results are included in the Supplementary Fig. 7 of the revised manuscript.

Figure 4. Photoinhibition of the ACC-BLA projection does not alter avoidance response to fox urine.

1d. Inhibits reactions during exposure to a live predator (a rat, for example).

To examine this issue, we performed new experiment where mice were exposed to a rat instead of fox urine while ACC-BLA projection was optogenetically inhibited as in the case of fox urine experiment (Figure 5). For the rat exposure test (RET), we referred to a previously reported protocol (Yang et al., 2004) with slight modifications (no amphetamine and no hiding shelter in our condition). In our RET condition, mice exhibited moderate level of freezing (approximately 20 % mean freezing). Similar to fox urine condition, here we found that freezing response was increased by photoinhibition of ACC-BLA projection in eNpHR3.0 group compared to eGFP control group (Figure 5). These results suggest that the control function of the ACC-BLA projection is not specific to the freezing response to the predator odor cue in the absence of a live predator. The predatory imminence theory posits that exposure to a visible live predator account for a state with increased imminence ('Post'-encounter stage), whereas exposure to only an odor induces low imminence ('Pre'-encounter stage) (Fanselow & Lester, 1988; LeDoux, 2015). Our results in this live rat test therefore suggest that control of freezing by ACC-BLA projection to a predator odor is not specific to low imminence threat situation. Despite this result, we feel that a lot more experiments are needed to make a solid conclusion whether the control function of ACC-BLA projection is specific to the odor (in the absence of a real predator) or it is also involved in the control of fear response to a live predator. Thus, we decided not to include these results in the revised manuscript.

Figure 5. Photoinhibition of the ACC-BLA projection induces increased freezing response under exposure to a live predator.

2. Investigate the specificity of this manipulation by testing if prelimbic cortex input to the BLA changes freezing to fox urine.

This is another important control experiment suggested by the reviewer that can address the specificity of our manipulation. Because prelimbic cortex (PL) is located close to the ACC and also has a projection to the BLA (McDonald, Mascagni, & Guo, 1996; Sesack, Deutch, Roth, & Bunney, 1989), it is possible that our optogenetic manipulation effect was caused by unintended manipulation of PL-BLA projection. To address this issue, as the reviewer suggested, we examined whether photoinhibition of the PL-BLA projection induces the increase of freezing behavior to fox urine. The same eNpHR3.0 construct used for ACC experiments was virally expressed in the PL area by using a different coordinate corresponding to the PL via AAV delivery. The coordinate for fiber-optic implant was identical to the ACC-BLA experiments (A/P -1.6 mm, M/L \pm 3.35 mm, D/V -4.5 mm from bregma). In this new experiment, we also included ACC-NpHR and ACC-eGFP groups as a control for the comparison purpose. We again found that freezing to fox urine was increased in mice with the photoinhibition of ACC-BLA projection but not with the photoinhibition of PL-BLA projection compared to the eGFP control. These results therefore further confirm the specificity of our manipulation of ACC-BLA projection. Of note is that there was a tendency of decrease in freezing by the photoinhibition of PL-BLA projection. These results are included in the Fig. 6i and 6j of the revised manuscript.

Figure 6. Photoinhibition of the PL-BLA projection does not induce increased freezing response under fox urine exposure.

3. The patch-clamping characterization of the ACC-BLA projection in Fig. 4i should be expanded.

3a. The authors claim that "To confirm the direct glutamatergic nature of ACC-BLA synaptic

connections, we performed slice patch-clamp recording of BLA neurons". However their measures do not demonstrate monosynaptic connectivity. Repeat figure 4i in the presence of tetrodotoxin and 4-aminopyridine to show that the BLA light-induced response is monosynaptic.

The reviewer correctly pointed out that our data in Fig. 4i of the original manuscript may not demonstrate the monosynaptic connectivity between the ACC and BLA. As the reviewer suggested, we carefully re-performed the slice patch recording in the presence of tetrodotoxin (TTX) and 4-aminopyridine (4-AP). In addition to this approach, we adopted Cre-dependent anterograde trans-neuronal labeling strategy by using AAV1-hSyn-Cre virus as shown in a recently published paper (Zingg et al., 2017), which allowed us to visually identify BLA neurons receiving ACC inputs likely in a monosynaptic fashion (Zingg et al., 2017). We injected two different viral constructs (AAV1-Cre and AAV2-DIO-ChR2-EYFP) in the ACC of a transgenic mouse line containing floxed-stop tdTomato (Figure 7a below). In this experimental design, ChR2 is expressed in the cells co-expressing Cre in the ACC, and their postsynaptic target neurons, if any, would be labeled as tdTomato+ cells within the BLA due to the anterograde transfer of AAV1-hSyn-Cre. Four weeks after the viral injection, indeed, we observed tdTomato-positive cells within the BLA (Figure 7b). In the slice recording, we patched such tdTomato-positive neurons and recorded postsynaptic potential response (PSP) by optogenetically stimulating ChR2-expressing ACC projection in the presence of TTX and 4-AP. Neurons recorded within the BLA area (10 of 11, 2 mice, 91%) showed light-evoked EPSP responses in this recording setup, which were almost completely abolished by the additional incubation with NBQX + APV (Figure 7c). These recording data thus support our conclusion that ACC projects to the BLA through the monosynaptic excitatory synapse. We replaced previous figures with these new results (Fig. 4l-n) in the revised manuscript.

Figure 7. Electrophysiological characterization of the ACC-BLA projection based on monosynaptic trans-neuronal labelling strategy.

Notably, when we examined the distribution of tdTomato-expressing cells in the amygdala in the brain slice used for the recording under the fluorescence microscope, tdTomato-positive cells were exclusively detected in the BLA, but not in other amygdala sub-nuclei including LA, BMA and CeA (Figure 7b above). In particular, we observed no tdTomato-positive cells in the CeA in this condition (4-week expression). To further investigate the possibility of a direct ACC projection to the CeA, we performed the same tdTomato imaging analysis by using two additional mice. Because it was possible that 4-week was too short for the efficient trans-neuronal labeling in the CeA, this time we waited 8 weeks before conducting the imaging analysis. We collected consecutive brain sections covering the entire BLA and CeA area (AP coordinate -1.0 to -1.9 mm), and compared the number of tdTomato+ cells in the BLA and CeA at each slide (Figure 8 below). Consistently, we again found a predominant expression of tdTomato+ cells in the BLA from two different mice (total 204 in mouse 1 and 77 cells in mouse 2, respectively). In the CeA, very few tdTomato+ cells were detected (total 8 in mouse 1 and 3 cells in mouse 2, respectively). Considering the possibility of di-synaptic delivery of AAV1-Cre (Zingg et al., 2017), we conclude that there is no or a very sparse ACC projection to the CeA. These results are included in the Supplementary Fig. 4 of the revised manuscript.

Figure 8. Tracing from the ACC using AAV1-hSyn-Cre reveals trans-neuronal labelling of cells in the BLA, but no or very few cells in the CeA.

3b. Show BLA cell responses to ACC terminal optical stimulation with the same parameters used in the behavioral experiments.

In order to address this concern raised by the reviewer, we performed additional slice patch recording experiments using another group of mice. First, we injected the same ChR2 virus used for behavioral experiments (AAV2/1 serotype, CaMKII α -ChR2-Venus, titer of 6×10^{11} viral genome/ml). Second, chemical antagonists (TTX and 4-AP) were not applied during the recording. Third, 10-Hz light pulses with 10-ms width, the optical stimulation condition used for behavioral experiments, were applied. Due to the technical limit, we used a lower light intensity (4 mW/mm^2 at the microscopic imaging plane) than the one (9.6 mW/mm^2 for ACC somata, 25 mW/mm^2 for ACC-BLA terminal, at the tip) used for behavioral experiments. Upon such 10-Hz light stimulation, the BLA cells (two mice) showed stable EPSP responses with a relatively high fidelity ($\sim 80\%$): in response to 10 blue light stimuli with the 10 Hz frequency, 4 of 5 BLA cells showed 10 EPSP responses (Figure 9), but only 1 of 5 cells showed 3 responses. By contrast, we found that none of the CeA cells (0 of 6 cells) in the same brain slice showed postsynaptic responses elicited by either single or 10 Hz optical stimulation (Figure 9), consistent with the observation with *tdTomato*-expressing brain slices described above. These recording results confirm the efficient manipulation of ACC-BLA projection by optogenetic approach in our condition and support our finding of no functional ACC input to the CeA. We included these results in the Supplementary Fig. 6 of the revised manuscript.

Figure 9. Representative post-synaptic potential responses to the stimulation of ACC-BLA pathway, tested with in-vivo-like parameters measured in BLA or CeA cells in acute brain slice.

4. *The authors should add a sentence or two discussing what, in their view, is the functional difference between the projections of the IL and ACC to the amygdala.*

It is unclear at this time how the ACC-BLA projection is functionally different from the IL-amygdala projection in the control of fear response. It could be that they may have a separate role for the inhibitory control of innate (ACC) versus learned (IL) fear response. Alternatively, ACC may have a cognitive control function for the fear response while IL is involved in the expression of fear extinction memory.

Minor points.

1. *Please provide the titer of CAV injections*

The titer of CAV virus was 6×10^{12} viral genome/ml. We reported this information in the Methods and Materials of the revised manuscript.

2. *The authors should remove the text " Thus, our circuit analyses indicate that the BLA is the major ACC projection target in the amygdala and receives direct glutamatergic excitatory input". Their data do not support this claim, as the authors did not demonstrate that the BLA received stronger ACC input than other amygdala nuclei. To support this claim the authors would have to systematically compare the strength of ACC input to various amygdala nuclei*

In response to the reviewer's point, we performed quantification analysis of fluorescence intensity of eGFP-Syb2 signals across different amygdala sub-regions by using new group of mice (3 mice) injected with eGFP-Syb2 in the bilateral ACC. Mean intensity values (fluorescence / mm^2 , 4 brain sections per mice) measured in the LA, BLA, CeL/C, CeM, and BLA were normalized to the value obtained from the LA (within each animal), an amygdala sub-region with no or little ACC input (Cassell & Wright, 1986; McDonald et al., 1996). The quantification analysis revealed that BLA region displayed the highest density within the amygdala sub-regions, consistently across 3 different mice (Figure 10 below). This finding supports that the BLA is the major ACC projection target in the amygdala. We added this result in the Fig. 4c of the revised manuscript.

Figure 10. Quantification of the eGFP-Syb2 labelled signals of the ACC projection to the amygdala sub-nuclei. (normalized to mean intensity value of the LA in each mouse)

Other notes.

The statistical analysis used was appropriate.

Reviewer #2

1. *The authors used an elegant viral strategy to presynaptically visualize ACC neurons the axonal projection in the amygdala, the authors claim the ACC is mainly projecting to the BLA and not to the central amygdala (CeL, CeC and CeM). They show a good example of this in Figure 4b, but there is no quantification across the different subfields of the amygdala (fluorescence / mm²) and even more importantly in different mice to identify how reproducible this observation is (see figure 3 and S3 of Beyeler et al., 2016). My concerns are even strengthened by the fluorescence image in Figure 4h where you can see almost more fluorescence in the central amygdala than in the BLA.*

We agree to the reviewer’s point that systematic quantification analysis is necessary to support our claim that the ACC is mainly projecting to the BLA. First, we performed quantification analysis of fluorescence intensity of eGFP-Syb2 signals across different amygdala sub-regions by using new group of mice (3 mice) injected with eGFP-Syb2 in the bilateral ACC. Mean intensity values (fluorescence / mm², 4 brain sections per mice) measured in the LA, BLA, CeL/C, CeM, and BLA were normalized to the value obtained from the LA, an amygdala sub-region with no or little ACC input (Cassell & Wright, 1986; McDonald et al., 1996). The quantification analysis revealed that BLA region displayed the highest density within the amygdala sub-regions (Figure 10), suggesting that the BLA is the major ACC projection target in the amygdala. Importantly, we confirm that these results were consistent across 3 different mice. We added this result in the Fig. 4c of the revised manuscript.

Notably, our eGFP-Syb2 fluorescence analysis is consistent with the results of a comparable tracing experiment (Figure 11 below) reported in the Allen connectivity atlas (experiment 478376911, AAV-Synaptophysin(Syp)-eGFP injection into the ACC, Figure 11).

Figure 10. Quantification of the eGFP-Syb2 labelled signals of the ACC projection to the amygdala sub-nuclei. (normalized to mean intensity value of the LA in each mouse)

Figure 11. Comparative intensity of the Syp-eGFP signals of the ACC projection to the amygdala sub-nuclei (experiment 478376911, available in the allen connectivity atlas, <http://connectivity.brain-map.org/>).

Second, in order to compare the ACC projection pattern across different amygdala sub-regions, this time we adopted Cre-dependent anterograde transneuronal labeling strategy by using AAV1-hSyn-Cre virus as shown in a recently published paper (Zingg et al., 2017), which allowed us to visually identify amygdala neurons receiving ACC input likely in a monosynaptic fashion (Zingg et al., 2017). As we described above, we injected two different viral constructs (AAV1-hSyn-Cre and AAV2-DIO-ChR2-EYFP) in the ACC of a transgenic mouse line containing floxed-stop tdTomato (Figure 7a above). In this experimental design, ChR2 is expressed in the cells co-expressing Cre in the ACC and their postsynaptic target neurons, if any, would be labeled as a tdTomato-positive cells due to the anterograde transfer of AAV1-hSyn-Cre. Four weeks after the viral injection, indeed, we observed tdTomato-positive cells in the amygdala (Figure 7b above). Importantly, tdTomato-positive cells were exclusively detected in the BLA, but not in other amygdala sub-nuclei including LA, BMA and CeA (Figure 6b). In particular, there were no tdTomato-positive cells detected in the CeA in this condition. We patched such tdTomato-positive BLA cells and recorded postsynaptic potential response (PSP) by optogenetically stimulating ChR-expressing ACC projection in the presence of TTX and 4-AP to test whether light-induced PSP is monosynaptic. Neurons recorded within the BLA area (10 of 11; 2 mice, 91%) showed light-evoked EPSP responses in this recording setup, which was almost completely abolished by the additional incubation with NBQX + APV (Figure 7c), demonstrating a monosynaptic excitatory connectivity of ACC-BLA projection. Furthermore, we also performed new patch recording of CeA cells by using mice injected with ChR2 in the ACC. All recorded CeA cells failed to show the light-evoked PSP response (total n=11; 2 mice). Because it was possible that 4-week was too short for the efficient trans-neuronal labeling in the CeA, this time we waited 8 weeks before conducting the imaging analysis. We collected consecutive brain sections covering the entire BLA and CeA area (AP coordinate -1.0 to -1.9 mm), and compared the number of tdTomato+ cells in the BLA and CeA at each slide (Figure 8 above). Consistently, we again found a predominant expression of tdTomato+ cells in the BLA from two different mice (total 204 in mouse 1 and 77 cells in mouse 2, respectively). In the CeA, this time very few tdTomato+ cells were detected (total 8 in mouse 1 and 3 cells in mouse 2, respectively). Considering the possibility of di-synaptic delivery of AAV1-Cre (Zingg et al., 2017), we conclude that there is no or a very sparse ACC projection to the CeA. Taken together, our analyses of eGFP-Syb2 fluorescence intensity, AAV1-Cre-mediated anterograde transneuronal labeling, and slice patch clamp recording data strongly support that the BLA is the major ACC projection target in the amygdala. We added these important data in the Supplementary Fig. 4 of the revised manuscript.

The reviewer also pointed out that in the Figure 4h of the originally submitted manuscript there are more fluorescence signals in the CeA than in the BLA. The image was taken under a bright field setting at a low magnification in order to indicate electrode position in the slice. The round spotlight (white circle) over CeA-BLA region was a ‘reflection’ of the illumination but not fluorescence signal. We excluded this image in the revised manuscript.

2. *The ACC also projects to many other downstream regions involved in fear-related behaviors such as, the mPFC or the ventral periaqueductal grey or the hippocampus. The eGFP-Syb2 fluorescence should also be quantified in these different regions and compared to the BLA.*

We agree to the reviewer's point that the brain-wide projection patterns might provide useful information for the comprehensive understanding of ACC-mediated top-down control of fear behaviors. However, because our study aims to investigate the potential role of ACC projection to the amygdala, we think that brain-wide projection pattern analysis is beyond our focus. Nevertheless, a set of tracing results presented in the Allen connectivity atlas shows the comparative intensity of ACC projection terminals, as represented by the Syp-eGFP signals (Experiment 478376911, Figure 12 below). According to the quantification data, brain regions including the prelimbic cortex (PL), dorsomedial thalamus (DMT) and periaqueductal gray (PAG) display even stronger intensities than BLA. Thus, it is possible that these ACC projections may also be involved in the control of innate fear behaviors.

Figure 12. Comparative intensity of the Syp-eGFP signals of the ACC projection to other fear-related brain regions.

(experiment 478376911, available in the allen connectivity atlas, <http://connectivity.brain-map.org/>)

We added these points in the discussion of the revised manuscript.

- Regarding the projection specific experiments, the authors injected 1000 nL of CAV2-cre virus in the BLA, which is a huge volume for an injection in the mouse, and questions the specificity of the staining. Indeed, 300 nL (less than 1/3) is the amount used in many studies in the field performing dual virus targeting of projection neurons. Again, the results of the anatomy of ACC-BLA projection experiments remain mainly qualitative. The author mention most of the projectors are ipsilateral, but do not provide counts of cell bodies in the ipsilateral vs contralateral part, and how this is consistent across animals.

The reviewer pointed out that CAV2-cre virus injection is less likely to be specific to the BLA because we injected larger volume (1000 nL) than generally used in many other studies (300 nL). Following the reviewer's recommendation, we repeated the CAV2-Cre-related experiments with 300 nL volume. First, we performed tracing experiment. Consistent with our previous results (1000 nL condition), we again found that cells in layer V of the ipsilateral ACC were densely labeled with eYFP expression. Importantly, such expression pattern was consistent across 5 different mice. To compare ipsilateral versus contralateral projection, we measured the average density of eYFP-positive cells in the ACC (4 brain sections per mouse, total 5 mice), as the reviewer suggested. This quantification analysis confirms our conclusion that ACC-BLA projection is mainly ipsilateral (Figure 13). We replaced previous data with these new data, which is now presented in the Fig 4e and 4 f of the revised manuscript.

Figure 13. Quantification of the retrogradely labelled BLA-projecting neurons in the ACC.

In addition, we also repeated the DIO-ChR2 behavior experiment (stimulating cell body of

ACC neurons projecting to BLA) by using the 300 nL volume of CAV2-Cre virus. We obtained consistent results (Figure 14 below). We replaced previous figures with these new results, which is now presented in the Fig. 5 of the revised manuscript.

4. *In the electrophysiological recordings the number of neurons is rather low (n=6) and the claims are not supported by the data. Indeed, in the abstract the author say “Circuit tracing and electrophysiological recordings identified a monosynaptic glutamatergic input from a subpopulation of ACC neurons to the BLA”. However, to demonstrate that a connection is monosynaptic, the author need to block polysynaptic activation in the slice, which is generally achieved by application of the sodium channel blocker TTX (see Figure 1 of Petreanu et al., 2009). The authors should also quantify the latency of onset of the photocurrent, which could be an indication of mono or polysynaptic transmission.*

The reviewer pointed out that the number of neurons recorded in the BLA in our slice patch recording is low. In response to the reviewer’s concern, we repeated same patch recording in the BLA cells to obtain more data (total n = 12; 2 mice). Moreover, this time we also recorded CeA cells in the same brain slice to examine light-evoked response as we also described in our response to the question #1. None of the CeA cells we recorded showed response (total n = 11; 2 mice). We added these additional recording results in the revised manuscript.

The reviewer correctly pointed out that our recording condition without TTX and 4-AP does not demonstrate a monosynaptic connectivity of ACC-BLA projection. For our response to this question, please refer to our response to the question #1 above. Overall latency of onset of the light-induced EPSPs was in the range of several millisecond (4.53 ± 0.95 ms; in the range of 1.9 ~ 10.5 ms), again consistent with that of excitatory monosynaptic responses.

5. *For the optogenetic inhibition of ACC terminals in the BLA, despite the authors mentioned the strongest inputs in the amygdala is in the BLA, they place the optic fibers in the central amygdala (Supplementary Figure 5). Similarly, to photoactivate the BLA-ACC neurons, the author placed the optic fibers in the central amygdala (Supplementary Figure 6). This compromises the interpretation of all the results of figure 6 and 7.*

We agree to the reviewer’s point that the placement of optic fiber is not specific to the BLA such that our optogenetic manipulation could affect ACC-CeA projection as well, although our data from tracing and slice recording suggest that no or very sparse, if any, ACC

projection to the CeA. To address the reviewer's concern, we repeated same eNpHR3.0 experiment but this time with different coordinate (-1.6 AP, ± 3.35 ML, -4.5 DV, mm from bregma) for the optic fiber placement corresponding to the BLA. Consistent with previous results, here we also found that inhibiting ACC projection significantly increased freezing response to the fox urine (Figure 15 below). These results thus further confirm our conclusion that ACC-BLA projection controls innate fear response. We added these important data in the Fig. 6g-j of the revised manuscript.

Figure 15. Photoinhibition of ACC-BLA projection induces increased freezing response under exposure to fox urine (re-performed due to the coordinate issue).

The reviewer also correctly pointed out similar issue in our WGA-Cre experiment. In response to the reviewer #3' recommendation (question #4), we excluded those data in the revised manuscript.

6. *It is unclear whether ACC inhibition during conditioned and innate fear stimuli was performed in the same experimental group. From the number of mice in the control group (7 for CS and 8 for US) it seems that the mice are from different experimental groups. Because, optogenetic manipulations are depending on many variables (location of viral injection, viral expression, fiber placement..) it would have been a good control to have the manipulations for both the CS and US in the same animals. If this is not the case, please make it clear.*

We agree with the reviewer that it would be ideal if we can test the ACC manipulation effect on both conditioned and innate freezing in the same animals. However, we tried to avoid any potential interaction between those two different experiments. For example, prior experience of shock may affect fear response to the fox urine vice versa. So, we used different groups of mice for conditioned and innate fear. We clarified this point in the result section and figure legend of the revised manuscript.

7. *The statistical comparison is not well described and seems not appropriate. Indeed, the authors*

use multiple paired t-tests for repeated measure of freezing across CS presentations (Figure 1d-e, j-k) where an ANOVA would be more suitable. Moreover, the same data is used in the differential freezing score, therefore, the p values should be corrected for multiple comparison.

Because the main purpose we repeated light OFF-ON session three times was to examine reproducibility but not time-dependent effect, we used paired t-test instead of ANOVA for the statistical analysis. Notably, we observed that freezing level was stable in GFP control group over time. As far as we understand, both paired t-test and ANOVA could be used to determine statistical significance in this case. As the reviewer commented, we also analyzed our data with a repeated measures (RM) one way ANOVA method and found consistent statistical significance for all data. For multiple comparison analysis, we used pair-wise Sidak's post-hoc test.

In addition to the reviewer's point, we carefully re-examined and corrected our use of parametric and non-parametric tests by conducting the Shapiro-Wilk normality test, and non-parametric tests were used for conditions where the normality test failed. Statistical significance was unchanged for all data. We improved the report of statistical analysis in the Methods section of the revised manuscript.

_____Minor comments_____

1. *The y-axis for the freezing to the conditioned cue is at 100% (Figure 1d-e) whereas the y-axis is at 60 % for the rest of the figure. This is misleading and suggest the authors want to reduce the perception of the impact of the optogenetic inhibition on the behavioral response to the conditioned cues.*

In response to the reviewer's comment, we made the y-axis 60% in all figures showing freezing in the revised manuscript.

2. *For optogenetic activation the author used blue light (473 nm wavelength, 0.2 mW at fiber tip). The light power seems very low in comparison with previous publications.*

The reviewer pointed out that 473 nm light power we used (0.2 mW at 200-um diameter tip corresponds to 9.6 mW/mm² in our condition) seems to be low. Similar issue was raised by the reviewer #1 (question 3b). In response to this concern, we performed additional slice patch recording in the BLA cells. We recorded BLA cells' response to the 10 Hz light stimulation (a parameter used for the behavioral experiments) of ChR2-expressing ACC projection. In this experiment, due to the technical limit, we used 4 mW/mm² light power. Even with lower light power, optical stimulation produced EPSP response with high fidelity in the BLA cells, confirming the efficient optogenetic manipulation of ACC-BLA projection in our condition.

3. *This sentence in the introduction is too long: Previous studies in rodents extensively investigated the role of prelimbic (PL) and infralimbic (IL) regions of the mPFC, which are homologous to the ventral-medial prefrontal cortex (vmPFC) and ventral anterior cingulate cortex (vACC) in humans^{1,3}, and their projections in the regulation of fear behavior in response to learned threat.*

To make this sentence short and clear, we divided it into two sentences as below in the revised manuscript. The prelimbic (PL) and infralimbic (IL) regions of the mPFC in rodent brains are homologous to the ventral-medial prefrontal cortex (vmPFC) and ventral anterior cingulate cortex (vACC) in humans^{1,3}. Previous studies in rodents extensively investigated the role of those two brain regions and their projections in the regulation of fear response to learned threat.

4. *Figure 4c: BLA: CAV2-cre should be above the BLA diagram/needle*

As the reviewer suggested, we corrected it in the revised manuscript. .

5. *The titer of the CAV2-cre virus should be specified.*

We reported the titer of the CAV2-cre virus (6×10^{12} viral genome/ml) in the methods section in the revised manuscript.

Reviewer #3

1. *First of all, the main finding of the study is quite surprising as the inhibition of the ACC-BLA pathway increases innate fear behavior whereas its activation decreases it. This suggests that inhibition of the ACC-BLA pathway is instrumental for innate fear behavior expression and that ACC local inhibitory interneurons are involved (activated) during innate fear behavior. It also suggests that under basal conditions the ACC-BLA pathway is only partially inhibited which lead to some innate fear behavior. I think this aspect deserve additional experimental work and will strengthen the impact of the manuscript. The authors could for instance perform some interneuronal recordings/imaging experiments during behavioral training (photometry, calcium imaging, single unit recordings).*

This is an insightful idea from the reviewer. Although the interneuron-dependent mechanism is a possible idea, we first investigated whether ACC cells projecting to the BLA is inactivated in the fox urine exposure condition before performing the interneuronal recordings/imaging experiments suggested by the reviewer. For this purpose, we took a c-Fos staining method to determine the activation of ACC cells projecting to the BLA. The ACC cells projecting to the BLA were labeled as eYFP+ cells by using dual viral injections of AAV-DIO-eYFP in the bilateral ACC and the retrograde CAV2-Cre virus in the bilateral BLA (Figure 16). Different groups of mice were exposed to either water (control) or fox urine, and then perfused 90 min later for c-Fos immunostaining. Interestingly, we found that the % number of eYFP and c-Fos double-positive cells among eYFP+ cells was not decreased, but rather increased in fox urine exposure condition compared to the water control condition. Our observation thus suggests that the activity of the ACC-BLA projection cells was increased while mice were exposed to the fox urine odor, which seems to be inconsistent with the above idea. Based on our c-Fos data, one possible hypothesis is that the activity of ACC-BLA projection is low at no threat condition but induced upon encountering innate threats to control the proper level of freezing response likely by activating freezing-inhibiting cells in the BLA. Although this is an important point, we need a lot more data to provide a clear answer to this issue. We believe that such works are beyond the purpose of this study and can be done in the future study.

- Upon light activation of the ACC alone or the ACC-BLA pathway, the authors observed a reduction in innate fear behavior. However, this drop in freezing behavior could also be related to non specific motor effects, which are not controlled in the present study.

The reviewer correctly pointed out that control data for the possibility of non-specific motor effects by ACC or ACC-BLA projection manipulation were missing in our study. We addressed this issue by performing new experiments where we repeated behavioral experiments with optical activation or inactivation of ACC-BLA projection by using optic fiber placement adjusted toward BLA (please see the question 5 of the reviewer #2 and the question 1b of the reviewer #1). Two days after fear response test, we performed open field test by using the same animals. We found no effects of our optogenetic manipulation on locomotor activity and anxiety-related behavior. These results further strengthen our conclusion. We included these important control data in the revised manuscript.

- It is largely assumed that fox odor induces an aversive/fear state, yet to my knowledge there is no studies demonstrating that the so called freezing observed upon fox odor exposure correspond really to freezing behavior and not just immobility. These two options could be disentangle using EMG recordings like in Steenland and Zhuo (2009).

To our understanding, this is a fundamental question of how fear state can be determined in mice raised by the reviewer. Although we agree with the reviewer that EMG recording could report fear state perhaps by distinguishing freezing from immobility, it was, however, not very convincing to us to use EMG value to define innate freezing. So, we took a different approach. We analyzed the activity of the dorsal periaqueductal gray (dPAG) in mice exposed to either fox urine or water as a control by using c-Fos immunostaining method. Evidence from cumulative research indicates that dPAG is a critical brain structure that evokes defensive behaviors including freezing while animals are exposed to innate threats such as predator-related threats or an actual predator (Gross & Canteras, 2012). It has been shown that dPAG is activated in response to the innate threats and its function is necessary for producing defensive behaviors including freezing (Cezario, Ribeiro-Barbosa,

Baldo, & Canteras, 2008; Sukikara, Mota-Ortiz, Baldo, Felicio, & Canteras, 2010). Moreover, a direct activation of dPAG is sufficient for inducing freezing (Bittencourt, Carobrez, Zamprogno, Tufik, & Schenberg, 2004). We found that c-Fos signal was increased specifically in the dPAG but not in the ventral PAG in mice exposed to the fox urine compared to water control (Figure 17 below). Four brain sections per mouse (n = 5 and n = 6 mice for water and fox urine condition, respectively) were used for c-Fos analysis. These results thus support that fox urine induces innate fear responses including freezing in mice and that the immobility behavior induced in fox urine exposure reflects fear-related freezing. We included these c-Fos data in the Supplementary Fig. 2 of the revised manuscript.

- The last section of the paper is out of focus of the present research and should be removed from the main manuscript. It raises additional questions and the rationale for performing this experiment is absolutely not clear.

We agree with the reviewer and excluded these results in the revised manuscript.

- The results of the present manuscript are quite contradictory with the work from Bissiere et al., *Biol Psy*, 2008, which partially target the caudal ACC. This work should be discussed.

The reviewer pointed out that our results are contradictory with the work from Bissiere et al., *Biol Psy*, 2008. Bissiere et al. investigated the effect of transient activation or inactivation of the rostral part of ACC (rACC) by using drug on the acquisition of auditory conditioned fear in rats. Their results show that the inactivation of rACC decreased conditioned freezing measured during training. Conversely, activation increased conditioned freezing response. These data seem to be contradictory with our results. One possible explanation is that ACC may have a different regulatory role for the expression of conditioned fear during training and during test 24 hr after training. Indeed, the identical inactivation of rACC 24 hr after training does not affect the expression of conditioned fear, which is consistent with our NpHR result. Alternative explanation is that the inconsistency may result from the difference in the ACC area manipulated in two studies. We targeted a more caudal part of the ACC but the study by Bissiere et al. targeted rostral ACC. In this sense, we speculate that caudal ACC may have an inhibitory control function for the

expression of conditioned fear and rostral ACC regulates the acquisition of conditioned fear perhaps by contributing to the processing of pain information during training. We included these explanations in the discussion section of the revised manuscript.

References

- Bittencourt, A. S., Carobrez, A. P., Zamprogno, L. P., Tufik, S., & Schenberg, L. C. (2004). Organization of single components of defensive behaviors within distinct columns of periaqueductal gray matter of the rat: role of N-methyl-D-aspartic acid glutamate receptors. *Neuroscience*, *125*(1), 71-89.
- Blanchard, D. C., Markham, C., Yang, M., Hubbard, D., Madarang, E., & Blanchard, R. J. (2003). Failure to produce conditioning with low-dose trimethylthiazoline or cat feces as unconditioned stimuli. *Behav. Neurosci.*, *117*(2), 360-368.
- Cassell, M., & Wright, D. (1986). Topography of projections from the medial prefrontal cortex to the amygdala in the rat. *Brain Res. Bull.*, *17*(3), 321-333.
- Cezario, A. F., Ribeiro-Barbosa, E. R., Baldo, M. V., & Canteras, N. S. (2008). Hypothalamic sites responding to predator threats--the role of the dorsal preammillary nucleus in unconditioned and conditioned antipredatory defensive behavior. *Eur. J. Neurosci.*, *28*(5), 1003-1015.
- Fanselow, M. S., & Lester, L. S. (1988). A functional behavioristic approach to aversively motivated behavior: Predatory imminence as a determinant of the topography of defensive behavior. *Evolution and Learning*, (Earlbaum, Hillsdale, New Jersey, 1988)
- Fendt, M. (2006). Exposure to urine of canids and felids, but not of herbivores, induces defensive behavior in laboratory rats. *J. Chem. Ecol.*, *32*(12), 2617-2627.
- Gross, C. T., & Canteras, N. S. (2012). The many paths to fear. *Nat Rev Neurosci*, *13*(9), 651-658.
- LeDoux, J. (2015). *Anxious: Using the brain to understand and treat fear and anxiety*. (Viking, New York City, New York, USA, 2015)
- McDonald, A., Mascagni, F., & Guo, L. (1996). Projections of the medial and lateral prefrontal cortices to the amygdala: a Phaseolus vulgaris leucoagglutinin study in the rat. *Neuroscience*, *71*(1), 55-75.
- Sesack, S. R., Deutch, A. Y., Roth, R. H., & Bunney, B. S. (1989). Topographical organization of the efferent projections of the medial prefrontal cortex in the rat: an anterograde tract-tracing study with Phaseolus vulgaris leucoagglutinin. *J. Comp. Neurol.*, *290*(2), 213-242.
- Sukikara, M. H., Mota-Ortiz, S. R., Baldo, M. V., Felicio, L. F., & Canteras, N. S. (2010). The periaqueductal gray and its potential role in maternal behavior inhibition in response to predatory threats. *Behav. Brain Res.*, *209*(2), 226-233.
- Wernecke, K. E., Vincenz, D., Storsberg, S., D'Hanis, W., Goldschmidt, J., & Fendt, M. (2015). Fox urine exposure induces avoidance behavior in rats and activates the amygdalar olfactory cortex. *Behav. Brain Res.*, *279*, 76-81.
- Yang, M., Augustsson, H., Markham, C. M., Hubbard, D. T., Webster, D., Wall, P. M., . . . Blanchard, D. C. (2004). The rat exposure test: a model of mouse defensive behaviors. *Physiol. Behav.*, *81*(3), 465-473.
- Zingg, B., Chou, X. L., Zhang, Z. G., Mesik, L., Liang, F., Tao, H. W., & Zhang, L. I. (2017). AAV-Mediated Anterograde Transsynaptic Tagging: Mapping Corticocollicular Input-Defined Neural Pathways for Defense Behaviors. *Neuron*, *93*(1), 33-47.

Reviewers' comments:

Reviewer #1 (Remarks to the Author):

Overall the authors have addressed all my concerns and I consider the paper acceptable for publication. There are two minor remaining issues:

1. The authors claim that their manipulation affected freezing elicited by coyote but not cat urine because cat urine doesn't provoke fear strongly. This contention is not supported by their data, as in the EGFP control group, both cat and coyote (and fox) urine provoke freezing at around 20%.
2. I was pleased to see the authors showed that their manipulations affected defensive behaviors elicited by a live predator (Rat). I agree with the authors that this experiment is not sufficient to prove that the acc-bla projection can mediate non-odor related freezing. However, I do not agree with the authors view that this data should be removed from the manuscript. Showing that their manipulation has consistent effects in mice exposed to a wide variety of predatorial stimuli further strengthens their claims and suggest the findings are robust, and as such I strongly suggest this data should be included in the paper.

Reviewer #2 (Remarks to the Author):

The authors considerably improved the study which still contains major, but addressable concerns regarding the results and their interpretation.

___Major Comments___

1. Abstract (Line 19): The authors write « Circuit tracing and slice patch recordings demonstrate a monosynaptic glutamatergic connectivity of ACC-BLA but no or very sparse ACC input to the central amygdala.”
When looking at the AAV-CaMKII-mCH-eYFP-Sby2 expressing, there is clearly expression in the other territories of the BAL. It is not true to say there is no inputs. Please replace with “Circuit tracing and slice patch recordings demonstrate a monosynaptic glutamatergic connection of ACC-BLA but very sparse ACC input to the central amygdala.”
2. The ex vivo physiology experiment still need to be improved to be publishable. Indeed, the population quantification need to appear on the main figure for
 - (1) the proportion of the neurons,
 - (2) the amplitude of the PSP in ACSF, Biccuculine and NBQX + APV, and
 - (3) the latencyThese analysis need to be performed for both experiments: recordings of unidentified BLA neurons and ACC input-identified BLA neurons (Figure 4l-n). Also, Figure 4j and 4n are missing raw PSPs traces (before Biccuculine, and before TTX+4AP application). Finally, part of Figure 4i-k and Figure 4l-n are redundant (i-k should be moved to the supplement).

Figure S6: The author mention fidelity, but this parameter classically refers to action potential fired for each stimulation. In their case they have 0% fidelity. To prevent confusion the authors should call it PSP-fidelity, or PSP-reliability. The figure should contain diagram with the proportion of cells responding or not and with the respective reliability. How many mCh(+) cells were recorded in the CeA ?
If the authors move Figure 4i-k to the supplementary section, the actual Figure 4 and 6 would be helpful in the main Figure 4.

Line 198: "PSPs were in an excitatory direction" Please remove "direction" which is misleading and unnecessary. Replace with "PSPs were excitatory".

Line 199: "enhanced PSPs". Please specify enhanced "amplitude" of the PSPs.

3. The author carefully show distribution of the fluorescence for one mouse AAV-eNpHR and AAV-ChR2, but do not show the center of injection of each animal, which should be added on the diagram on the left of the fluorescence images. Moreover, the location of the fiber tip is also not represented and is even not visible on the fluorescence images.

4. Placement of optic-fibers: Regarding the point 5 of the first review

5. For the optogenetic inhibition of ACC terminals in the BLA, despite the authors mentioned the strongest inputs in the amygdala is in the BLA, they place the optic fibers in the central amygdala (Supplementary Figure 5). Similarly, to photoactivate the BLA-ACC neurons, the author placed the optic fibers in the central amygdala (Supplementary Figure 6). This compromises the interpretation of all the results of figure 6 and 7. We agree to the reviewer's point that the placement of optic fiber is not specific to the BLA such that our optogenetic manipulation could affect ACC-CeA projection as well, although our data from tracing and slice recording suggest that no or very sparse, if any, ACC projection to the CeA. To address the reviewer's concern, we repeated same eNpHR3.0 experiment but this time with different coordinate (-1.6 AP, \pm 3.35 ML, -4.5 DV, mm from bregma) for the optic fiber placement corresponding to the BLA. Consistent with previous results, here we also found that inhibiting ACC projection significantly increased freezing response to the fox urine (Figure 15 below). These results thus further confirm our conclusion that ACC-BLA projection controls innate fear response. We added these important data in the Fig. 6g-j of the revised manuscript.

I appreciate that the authors repeated the experiment with the optic-fiber actually placed over the BLA. However, the author still maintain that they are targeting the ACC-BLA terminals with their placement at ML \pm 3.2 mm. This is not true and with the 3.2 mm fiber placements, the experimenters are inhibiting the sparse ACC-CeA projection. This is an interesting result that should be described as it is. The manuscript cannot be published with the statement that the 3.2 mm coordinate is targeting ACC fibers in the BLA.

5. While the authors controlled for locomotor and anxiety effect of the AAC-BLA activation, these important controls are missing for the ACC cell bodies activation and inhibition. Moreover this control can be represented in the supplementary figure in order to make Figure 6 more accessible.

____Minor Comments____

Title : The first 3 Figures of the study are focusing on the ACC cell bodies, therefore, I would advise that the authors also highlight this in their title : « Control of innate fear response to predator odor via anterior cingulate cortex AND ITS input to the basolateral amygdala in mice. »

Across the manuscript please use a more specific terminology to describe the electrophysiological experiments, i.e. « slice patch » = « ex vivo patch-clamp »

Introduction_____

Line 40: When the author summarize the De-Monte et al. study they oversimplify their findings. Indeed, (1) inactivation of PL does not abolish the conditioned freezing responses but decreases it from 80 to 40% 6h and 7d after conditioning

(2) inactivation of PL-BLA decreases it from 80 to 35% only 6h after, and

(3) inactivation of PL-CeA decreases it from 80 to 30% only 7 a days after.

It is important to rephrase this section.

Line 58 and 60: please specify the method of silencing in these studies

Line 64: replace "plays a control role" with "controls"

Results_____

Line 161: glutamatergic and synaptic are redundant. And monosynaptic is already redundant with synaptic. Please remove synaptic from the title.

Line 168: The description of the ACC terminals quantification is approximate and unclear. "AAV vector (AAV-eGFP-Syb2) containing eGFP fused-Syb2 with mCherry-IRES was injected into the bilateral ACC". First a vector is not containing but carrying the gene coding for a protein (here eGFP-Syb2). Second, the vector is also carrying mCherry-IRES and should therefore be noted AAV-eGFP-Syb2-mCh.

Reviewer #3 (Remarks to the Author):

The authors performed additional experiments to evaluate if the ACC-BLA pathway was inhibited or not during innate fear behavior. They show using c-fos immunostaining that ACC-BLA cells were rather excited upon fox urine exposure. Thus it is not clear to me how these findings reconcile with the main optogenetic effect of this study showing that inhibition of this pathway promotes innate fear behavior. Unless I missed something, there are some inconsistencies between the main findings and the present results. In the one hand, the ACC-BLA pathway is activated during innate fear, on the other hand, inhibition of this pathway promotes innate fear, which is exactly the opposite of what is expected. Some recordings of either the ACC or BLA should be provided to disentangle these discrepancies. In particular, these discrepancies could be related to the non-specific use of optogenetic approaches that are not intended to infer with the specific firing activity of ACC neurons projecting to the BLA. Again, recordings are required here.

Reviewers' comments:

Reviewer #1 (Remarks to the Author):

Overall the authors have addressed all my concerns and I consider the paper acceptable for publication. There are two minor remaining issues:

1. The authors claim that their manipulation affected freezing elicited by coyote but not cat urine because cat urine doesn't provoke fear strongly. This contention is not supported by their data, as in the EGFP control group, both cat and coyote (and fox) urine provoke freezing at around 20%.

The reviewer correctly pointed out that both cat and coyote urine provoked similar level of freezing at around 20% in our study. So, our results do not exclude the possibility that ACC-BLA projection does not control innate freezing response to cat urine. Nevertheless, because previous studies show that cat urine does not induce strong fear response in rodents (Blanchard et al., 2003; Fendt, 2006), it is highly likely that our optogenetic manipulation affected freezing elicited by coyote but not cat urine because cat urine doesn't provoke fear strongly. We added this point in the result of the revised manuscript.

2. I was pleased to see the authors showed that their manipulations affected defensive behaviors elicited by a live predator (Rat). I agree with the authors that this experiment is not sufficient to prove that the acc-bla projection can mediate non-odor related freezing. However, I do not agree with the authors view that this data should be removed from the manuscript. Showing that their manipulation has consistent effects in mice exposed to a wide variety of predatorial stimuli further strengthens their claims and suggest the findings are robust, and as such I strongly suggest this data should be included in the paper.

Following the reviewer's suggestion, we included the data in the revised manuscript.

Reviewer #2 (Remarks to the Author):

The authors considerably improved the study which still contains major, but addressable concerns regarding the results and their interpretation.

___Major Comments___

1. Abstract (Line 19): The authors write « Circuit tracing and slice patch recordings demonstrate a monosynaptic glutamatergic connectivity of ACC-BLA but no or very sparse ACC input to the central amygdala.» When looking at the AAV-CaMKII-mCH-eYFP-Sby2 expressing, there is clearly expression in the other territories of the BAL. It is not true to say there is no inputs. Please replace with “Circuit tracing and slice patch recordings demonstrate a monosynaptic glutamatergic connection of ACC-BLA but very sparse ACC input to the central amygdala.”

Following the reviewer's point, we replaced the sentence in the abstract of the revised manuscript.

2. The ex vivo physiology experiment still need to be improved to be publishable.

Indeed, the population quantification need to appear on the main figure for

(1) the proportion of the neurons,

(2) the amplitude of the PSP in ACSF, Biccuculine and NBQX + APV, and

(3) the latency

These analysis need to be performed for both experiments: recordings of unidentified BLA neurons and ACC input-identified BLA neurons (Figure 4l-n).

In order to improve the *ex vivo* electrophysiology experiment as the reviewer pointed out, we re-performed these experiments for both ACC input-identified and unidentified BLA neurons by using new sets of mice.

- ACC input-identified BLA neurons

To test whether ACC-BLA synapses are glutamatergic and monosynaptic, we measured the effects of sequential treatments of ACSF, bicuculine, bicuculine + TTX + 4-AP, and bicuculine + TTX + 4-AP + NBQX + APV on light-evoked postsynaptic potentials (PSPs). With acute BLA slices from tdTomato-floxed mice injected with AAV1-hSyn-Cre and AAV2-DIO-ChR2-EYFP in the ACC, we measured blue light (470nm, 1 ms)-evoked PSPs from tdTomato-labelled BLA cells. Of 17 BLA cells recorded from 10 slices of 5 mice, 12 BLA cells (71%) showed successful PSP responses but no detectable PSP responses (29%) in 5 cells. The latency of light-evoked PSP was 6.4 ± 0.7 ms (n=12), which is in the range of monosynaptic events. Among those 12 cells, we obtained successful recording data from 6 BLA cells with sequential drug treatments. Our data show that application of bicuculine slightly enhanced amplitudes of PSPs. These PSPs were not affected by additional TTX + 4-AP treatment, again confirming that light-evoked PSPs are mediated by the monosynaptic event. However, following co-application of NBQX and APV completely blocked PSP responses (ACSF vs. +NBQX, APV; n=6 cells from 6 slices of 3 mice; *p<0.05, repeated measures one-way ANOVA with Dunnett's multiple comparisons test). Of note is that we detected no tdTomato-labelled cells in the CeA in the same brain slices. . We replaced previous results with these new results, which are presented in the Figure 4 of the revised manuscript.

- Unidentified (randomly selected) BLA neurons

Same experiments were done with unidentified BLA or CeA cells. Randomly selected BLA or CeA cells in the slices from age-matched 129 x C57Bl/6 wild type mice injected with AAV-CaMKII α -ChR2-Venus in the ACC were recorded by the whole-cell patch clamp method: similar with the results from ACC input-identified BLA cells, 11 of total 14 BLA cells recorded from 14 slices of 5 mice showed successful light-evoked PSPs (79 %) with no detectable PSPs in 3 of 14 cells (21%). The latency of light-evoked PSPs was 7.6 ± 0.3 ms, which suggests a monosynaptic event. We obtained successful recordings from those 11 cells in testing the effects of sequential drug treatments on PSP amplitudes. Our results show that application of bicuculline causes the enhancement of PSP amplitudes, which were not altered by additional TTX and 4-AP. The PSPs were significantly inhibited by the addition of NBQX and APV (ACSF vs. other conditions; n=11 cells from 11 slices of 5 mice; *p<0.05, **p<0.01, repeated measures one-way ANOVA with Dunnett's multiple comparisons test). On the contrary, we could not detect any light-induced PSP responses from any of CeA cells (11 cells). We replaced previous results with these new results, which are presented in the supplementary Figure 5 of the revised manuscript.

Taken together, these results from new *ex vivo* electrophysiology experiments indicate that the ACC- BLA synapses are monosynaptic and glutamatergic.

Also, Figure 4j and 4n are missing raw PSPs traces (before Bicuculine, and before TTX+4AP application).

As the reviewer pointed out, we included raw PSPs traces in the revised manuscript.

Finally, part of Figure 4i-k and Figure 4l-n are redundant (i-k should be moved to the supplement).

As the reviewer suggested, Fig. 4i-k was moved to the supplementary Fig. 5 in the revised manuscript.

Figure S6: The author mention fidelity, but this parameter classically refers to action potential fired for each stimulation. In their case they have 0% fidelity. To prevent confusion the authors should call it PSP-fidelity, or PSP-reliability. The figure should contain diagram with the proportion of cells responding or not and with the respective reliability. How many mCh(+) cells were recorded in the CeA ?

As suggested by the reviewer, we changed “fidelity” to “PSP-reliability”. The diagram showing the proportion of cells displaying light (10Hz)-evoked PSPs was added to the supplementary Fig. 5 with the number of tested cells in the BLA and CeA.

For this recording experiment, we used 129 x C57Bl/6 wild-type mice, the one used for the behavior experiments, so there were no mCh(or tdTomato)(+) cells in the CeA. Moreover, in our recording experiments performed by using Ai9 transgenic mice, we observed no tdT+ cells in the CeA area so there was no tdT+ cells recorded in the CeA.

If the authors move Figure 4i-k to the supplementary section, the actual Figure 4 and 6 would be helpful in the main Figure 4.

As the reviewer recommended, we moved Fig. 4i-k (of previous version) to the supplementary Fig. 5 and the supplementary Fig. 6 (of previous version) to the main Fig. 5o-q in the revised manuscript.

Line 198: “PSPs were in an excitatory direction” Please remove “direction” which is misleading and unnecessary. Replace with “PSPs were excitatory”.

As the reviewer suggested, we changed “excitatory direction” into “excitatory” in the revised manuscript.

Line 199: “enhanced PSPs”. Please specify enhanced “amplitude” of the PSPs.

In response to the reviewer’s point, we changed “PSPs” into “the amplitude of the PSPs” in the revised manuscript.

3. The author carefully show distribution of the fluorescence for one mouse AAV-eNpHR and AAV-ChR2, but do not show the center of injection of each animal, which should be added on the diagram on the left of the fluorescence images. Moreover, the location of the fiber tip is also not represented and is even not visible on the fluorescence images.

As the reviewer suggested, we added the center of injection of each animal, and the location of the fiber tip in the supplementary Fig. 1 and 4 in the revised manuscript.

4. Placement of optic-fibers: Regarding the point 5 of the first review

5. For the optogenetic inhibition of ACC terminals in the BLA, despite the authors mentioned the strongest inputs in the amygdala is in the BLA, they place the optic fibers in the central amygdala (Supplementary Figure 5). Similarly, to photoactivate the BLA-ACC neurons, the author placed the optic fibers in the central amygdala (Supplementary Figure 6). This compromises the interpretation of all the results of figure 6 and 7. We agree to the reviewer's point that the placement of optic fiber is not specific to the BLA such that our optogenetic manipulation could affect ACC-CeA projection as well, although our data from tracing and slice recording suggest that no or very sparse, if any, ACC projection to the CeA. To address the reviewer's concern, we repeated same eNpHR3.0 experiment but this time with different coordinate (-1.6 AP, \pm 3.35 ML, -4.5 DV, mm from bregma) for the optic fiber placement corresponding to the BLA. Consistent with previous results, here we also found that inhibiting ACC projection significantly increased freezing response to the fox urine (Figure 15 below). These results thus further confirm our conclusion that ACC-BLA projection controls innate fear response. We added these important data in the Fig. 6g-j of the revised manuscript.

I appreciate that the authors repeated the experiment with the optic-fiber actually placed over the BLA. However, the author still maintain that they are targeting the ACC-BLA terminals with their placement at ML \pm 3.2 mm. This is not true and with the 3.2 mm fiber placements, the experimenters are inhibiting the sparse ACC-CeA projection. This is an interesting result that should be described as it is. The manuscript cannot be published with the statement that the 3.2 mm coordinate is targeting ACC fibers in the BLA.

In response to the reviewer's concern, we revised the result section and also added this point in the discussion section in the revised manuscript.

5. While the authors controlled for locomotor and anxiety effect of the AAC-BLA activation, these important controls are missing for the ACC cell bodies activation and inhibition. Moreover this control can be represented in the supplementary figure in order to make Figure 6 more accessible.

To address the reviewer's concern, we performed new open field test with ACC somata activation and inhibition. Again, we found no significant changes in basal locomotor activity and anxiety level with such optogenetic manipulation compared to the EGFP control. These important control results are presented in the supplementary Fig. 3 of the revised manuscript.

___Minor Comments___

Title : The first 3 Figures of the study are focusing on the ACC cell bodies, therefore, I would advise that the authors also highlight this in their title : « Control of innate fear response to predator odor via anterior cingulate cortex AND ITS input to the basolateral amygdala in mice. »

Following the reviewer's advice, we changed the title of our manuscript as “Control of innate fear response to predator odor via the anterior cingulate cortex and its input to the basolateral amygdala in mice”.

Across the manuscript please use a more specific terminology to describe the electrophysiological experiments, i.e. « slice patch » = « ex vivo patch-clamp »

As the reviewer suggested, we changed “slice patch” to “ex vivo (whole-cell) patch-clamp” or “ex vivo electrophysiology”. In addition, we carefully revised our manuscript to describe electrophysiological experiments with a more specific terminology as best as we can.

Introduction_____

Line 40: When the author summarize the De-Monte et al. study they oversimplify their findings. Indeed, (1) inactivation of PL does not abolish the conditioned freezing responses but decreases it from 80 to 40% 6h and 7d after conditioning

(2) inactivation of PL-BLA decreases it from 80 to 35% only 6h after, and

(3) inactivation of PL-CeA decreases it from 80 to 30% only 7 a days after.

It is important to rephrase this section.

In response to the reviewer's comment, we changed this section into as follows: "Conversely, the PL is thought to stimulate the expression of learned fear. A recent study in mice showed that inactivation of PL somata decreases the conditioned freezing response from 80 to 40% 6h and 7d after conditioning".

Line 58 and 60: please specify the method of silencing in these studies

In response to the reviewer's point, we specified the method of silencing (by lidocaine and optogenetics) in each study in the revised manuscript.

Line 64: replace "plays a control role" with "controls"

We replaced "plays a control role" with "controls" in the revised manuscript.

Results_____

Line 161: glutamatergic and synaptic are redundant. And monosynaptic is already redundant with synaptic. Please remove synaptic from the title.

The title sentence is changed into "ACC projection to the BLA is monosynaptic and glutamatergic" in the revised manuscript.

Line 168: The description of the ACC terminals quantification is approximate and unclear. "AAV vector (AAV-eGFP-Syb2) containing eGFP fused-Syb2 with mCherry-IRES was injected into the bilateral ACC". First a vector is not containing but carrying the gene coding for a protein (here eGFP-Syb2). Second, the vector is also carrying mCherry-IRES and should therefore be noted AAV-eGFP-Syb2-mCh.

Following the reviewer's suggestion, we changed this sentence as follows in the revised manuscript: "AAV vector (AAV-mCh-IRES-eGFP-Syb2) carrying the gene of eGFP fused-Syb2 with mCherry-IRES was injected into the bilateral ACC".

Reviewer #3 (Remarks to the Author):

The authors performed additional experiments to evaluate if the ACC-BLA pathway was inhibited or not during innate fear behavior. They show using c-fos immunostaining that ACC-BLA cells were rather excited upon fox urine exposure. Thus it is not clear to me how these findings reconcile with the main optogenetic effect of this study showing that inhibition of this pathway promotes innate fear behavior. Unless I missed something, there are some inconsistencies between the main findings and the present results. In the one hand, the ACC-BLA pathway is activated during innate fear, on the other hand, inhibition of this pathway promotes innate fear, which is exactly the opposite of what is

expected. Some recordings of either the ACC or BLA should be provided to disentangle these discrepancies. In particular, these discrepancies could be related to the non-specific use of optogenetic approaches that are not intended to infer with the specific firing activity of ACC neurons projecting to the BLA. Again, recordings are required here.

To make it clear, we do not hypothesize that ACC-BLA projection is activated upon exposure to the fox urine. Due to the lack of temporal specificity, our c-fos data do not exclude the possibility that the ACC cells were inactivated. Because mice showed alternate pattern of freezing and non-freezing behaviors in the presence of fox urine, it is possible that ACC neuronal activity also showed similar pattern.

So, in order to address the reviewer's concern, we performed two new experiments.

First, our results from c-fos immunostaining show that there is a basal activity in ACC-BLA pathway as shown in the water control group (about 10%). So, it is possible that local inhibitory interneuron in the ACC may be activated upon exposure to the fox urine to promote freezing response as the reviewer suggested in the first review comments. To test this possibility, we performed double immunostaining for c-fos and GAD67 (a marker for interneuron) in the water and fox urine condition. Consistent with the idea, we found that GAD67+ cells were activated in the presence of fox urine compared to the control, supporting that the inhibitory interneurons in the ACC are involved in innate fear expression. We included these important new data in the results section (main Fig. 1) and also added discussions in the revised manuscript.

Second, we have performed a pilot experiment for fiber photometry recording in the ACC where GCaMP6f was selectively expressed in the ACC neurons projecting to the BLA by using CAV2-Cre and AAV-DIO-GCaMP6f viral injection. While we are doing this experiment, we realized that it is technically almost impossible to determine how the activity of ACC-BLA projection is regulated by fox urine. First, it was impossible to determine exactly when the fox urine is perceived to mice in our condition because it is an odorant stimulus. If the threat is a shock or tone, it would be possible to know when the threat is delivered to the animals. Second, freezing response to fox urine is not instantly induced and continuously sustained in the presence of the fox urine rather the time duration animals show freezing is gradually increased over time. Thus, it was technically unfeasible for us to precisely link GCaMP6 activity to the induction of freezing. We agree with the reviewer that it is important to know how the ACC-BLA pathway responds to the fox urine but we believe that recording would not yield a meaningful outcome to make any clear conclusion at least in our current experimental conditions.

REVIEWERS' COMMENTS:

Reviewer #2 (Remarks to the Author):

The manuscript by Jhang et al. significantly improved again with the last revisions and is almost ready for publication. Some last, but necessary changes still need to be made before publication. This study of ACC-BLA projection in innate fear response is carefully performed, well presented and of high interest to the field.

Regarding the Figure 6 (and Supplementary Figure 6): the authors updated the manuscript depending on my comments, but some flows remain. Indeed, in Figure 6, the author write the coordinates, instead of writing the projection that was stimulated:

Fig4d-e: ACC-CeA,

Fig.4g-h and Fig.4l-m :ACC-BLA

(as it is already written for Fig.4i, IL-BLA.

Similarly, for figure 4j, the author CANNOT combine the data where the optic fiber is over the CeA and over the BLA and name it ACC-BLA.

Moving the ACC-CeA and IL-BLA data to the supplementary figures would strongly increase the clarity of this figure.

Finally, the fact that ACC-CeA inhibition is also increasing freezing is an important point which should be discussed in the discussion.

The ACC input within the BLA seems to be denser in the basal section of the BLA (BA) compared to the lateral section (LA) of the BLA. If this is the case, the authors should discuss this point and report if previous studies have identified this projection pattern.

In the discussion the authors extensively discuss the basomedial amygdala. The BMA sub-region of the BLA should be defined in the introduction. Especially, because the authors also quantify the quantity of terminals in this BLA sub-region (Fig. 4c).

When discussing the role of BLA-CeA projection, the authors should integrate the diversity of the CeA (CeL + CeM). Indeed, the CeL is sending feedforward inhibition to the CeM and both sub-regions are therefore supposed to have opposing roles. Indeed, BLA-CeM were shown to induce real-time place aversion and are preferentially excited to predictive cues of negative valence (Beyeler et al., 2016; Namburi et al., 2015).

The authors should discuss their result in perspective of recent findings on the involvement of ACC-BLA inputs in observational learning (Allsop et al., 2018).

The pharmacological agents for the ex vivo experiments should be better described.

Indeed, Bicuculline, AP5, NBQX TTX and 4AP are only described as inhibitors, which is not accurate.

Bicuculline is a GABAA antagonist, AP5 is a NMDA receptor antagonist, NBQX is an AMPA receptor antagonist, TTX is a sodium channel blocker, and 4-AP is a blocker of the voltage dependant potassium channels.

Also the authors suggest that "bicuculline application slightly enhanced PSP amplitudes". However, there is no statistical difference (see legend Fig4.n). The author should remove this sentence. Also the star indicating the significant difference should be written above a line going from ACSF to NBQX/AP5.

Also, Figure 4n would benefit of having each datapoint represented on top of the average (average could then be represented as histogram bars).

"These PSPs were completely disappeared" is not correct English. Please replace with These PSP were completely blocked.

Reviewer #3 (Remarks to the Author):

The authors in a new set of experiments demonstrated that GAD+ cells in the ACC are activated during innate freezing responses suggesting a direct inhibition of ACC-BLA projecting glutamatergic neurons.

Although these data clearly improve the manuscript, I will be more confident in the results if the authors could finally demonstrate that the photoactivation of ACC GAD+ cells promotes freezing behavior upon exposure to fox urine.

Reviewer #2 (Remarks to the Author):

The manuscript by Jhang et al. significantly improved again with the last revisions and is almost ready for publication. Some last, but necessary changes still need to be made before publication. This study of ACC-BLA projection in innate fear response is carefully performed, well presented and of high interest to the field.

Regarding the Figure 6 (and Supplementary Figure 6): the authors updated the manuscript depending on my comments, but some flows remain. Indeed, in Figure 6, the author write the coordinates, instead of writing the projection that was stimulated:

Fig4d-e: ACC-CeA,

Fig.4g-h and Fig.4l-m: ACC-BLA

(as it is already written for Fig.4i, IL-BLA.

Similarly, for figure 4j, the author CANNOT combine the data where the optic fiber is over the CeA and over the BLA and name it ACC-BLA.

Moving the ACC-CeA and IL-BLA data to the supplementary figures would strongly increase the clarity of this figure.

In response to the reviewer's points, we removed the coordinate and instead included 'ACC projection' or 'PL projection' in the revised figures. In addition, we moved Fig. 6d-f to the Supplementary Fig. 6 in the revised manuscript. We feel that naming ACC-CeA or ACC-BLA is incorrect because it is unclear whether light illumination in our condition was specific to either BLA or CeA. The PL projection experiment was performed together with ACC projection for the comparison purpose so we presented PL data in the main Figure 6 as before.

Finally, the fact that ACC-CeA inhibition is also increasing freezing is an important point which should be discussed in the discussion.

As we explained above, it was unclear whether our optical activation was specific to the ACC-CeA projection in the coordinate of ML \pm 3.2. Thus, it is possible that ACC projection to the CeA may have a contribution for the control of innate fear but our data are limited to discuss the specific role of ACC-CeA. We included this point in the discussion of our manuscript in the last revision.

The ACC input within the BLA seems to be denser in the basal section of the BLA (BA) compared to the lateral section (LA) of the BLA. If this is the case, the authors should discuss this point and report if previous studies have identified this projection pattern.

Previous tracing studies also reported similar ACC projection pattern with our data such that strong projection to the basolateral nucleus of amygdala (BLA) but not or very few to the lateral nucleus of amygdala (LA). We made this point clear in the result section of the revised manuscript. In addition, to avoid ambiguity, we clearly defined BLA as a basolateral nucleus of amygdala in the revised manuscript following a previous literature (Mouse brain atlas; Paxinos and Franklin, 2004).

In the discussion the authors extensively discuss the basomedial amygdala. The BMA sub-region of the BLA should be defined in the introduction. Especially, because the authors also quantify the quantity of terminals in this BLA sub-region (Fig. 4c).

Because we referred to Mouse brain atlas (Paxinos and Franklin, 2004) for the nomenclature of the sub-regions in the amygdala, we added the reference in the legend of Fig. 5 of the revised

manuscript where we defined BMA as a basomedial nucleus of amygdala.

When discussing the role of BLA-CeA projection, the authors should integrate the diversity of the CeA (CeL + CeM). Indeed, the CeL is sending feedforward inhibition to the CeM and both sub-regions are therefore supposed to have opposing roles. Indeed, BLA-CeM were shown to induce real-time place aversion and are preferentially excited to predictive cues of negative valence (Beyeler et al., 2016; Namburi et al., 2015).

We agree with the reviewer's point. We specified the sub-region within CeA (CeL or CeM) in the discussion of the revised manuscript.

The authors should discuss their result in perspective of recent findings on the involvement of ACC-BLA inputs in observational learning (Allsop et al., 2018).

This recent study shows a role of ACC-BLA circuit for encoding socially-derived aversive cue information in observational learning such that its inactivation during conditioning impairs the acquisition, but not expression, of observational fear conditioning. We added discussion for this finding in the revised manuscript.

The pharmacological agents for the ex vivo experiments should be better described. Indeed, Bicuculline, AP5, NBQX TTX and 4AP are only described as inhibitors, which is not accurate. Bicuculline is a GABAA antagonist, AP5 is a NMDA receptor antagonist, NBQX is an AMPA receptor antagonist, TTX is a sodium channel blocker, and 4-AP is a blocker of the voltage dependant potassium channels.

In response to the reviewer's point, we more accurately described the drugs in the method section of the revised manuscript.

Also the authors suggest that "bicuculline application slightly enhanced PSP amplitudes". However, there is no statistical difference (see legend Fig4.n). The author should remove this sentence.

We changed this sentence into "although there was no statistical significance, we observed a tendency that bicuculline application slightly enhanced PSP amplitudes".

Also the star indicating the significant difference should be written above a line going from ACSF to NBQX/AP5. Also, Figure 4n would benefit of having each data point represented on top of the average (average could then be represented as histogram bars).

In response to the reviewer's suggestion, we changed the main Fig. 5n and supplementary Fig. 5e in the revised manuscript.

"These PSPs were completely disappeared" is not correct English. Please replace with These PSP were completely blocked.

We replaced the sentence in the revised manuscript as the reviewer recommended.

Reviewer #3 (Remarks to the Author):

The authors in a new set of experiments demonstrated that GAD+ cells in the ACC are activated during innate freezing responses suggesting a direct inhibition of ACC-BLA projecting glutamatergic neurons.

Although these data clearly improve the manuscript, I will be more confident in the results if the authors could finally demonstrate that the photoactivation of ACC GAD+ cells promotes freezing behavior upon exposure to fox urine.

We extended our discussion on the potential mechanism underlying how ACC-BLA circuit controls innate fear response.

“Although the connectivity between the GABAergic interneurons (activated by innate threats) and ACC-BLA cells is unknown in our study, it is possible that such local inhibitory interneurons directly inhibits ACC-BLA cells to promote freezing response”.